# Simple Mechanisms for Welfare Maximization in Rich Advertising Auctions

**Gagan Aggarwal**
Google Research
gagana@google.com

**Kshipra Bhawalkar**
Google Research
kshipra@google.com

**Aranyak Mehta**
Google Research
aranyak@google.com

**Divyarthi Mohan**
Tel Aviv University
divyarthim@tau.ac.il

**Alexandros Psomas**
Purdue University
apsomas@cs.purdue.edu

## Abstract

Internet ad auctions have evolved from a few lines of text to richer informational layouts that include images, sitelinks, videos, etc. Ads in these new formats occupy varying amounts of space, and an advertiser can provide multiple formats, only one of which can be shown. The seller is now faced with a multi-parameter mechanism design problem. Computing an efficient allocation is computationally intractable, and therefore the standard Vickrey-Clarke-Groves (VCG) auction, while truthful and welfare-optimal, is impractical.

In this paper, we tackle a fundamental problem in the design of modern ad auctions. We adopt a "Myersonian" approach and study allocation rules that are monotone both in the bid and set of rich ads. We show that such rules can be paired with a payment function to give a truthful auction. Our main technical challenge is designing a monotone rule that yields a good approximation to the optimal welfare. Monotonicity doesn't hold for standard algorithms, e.g. the incremental bang-per-buck order, that give good approximations to "knapsack-like" problems such as ours. In fact, we show that no deterministic monotone rule can approximate the optimal welfare within a factor better than 2 (while there is a non-monotone FPTAS). Our main result is a new, simple, greedy and monotone allocation rule that guarantees a 3 approximation. In ad auctions in practice, monotone allocation rules are often paired with the so-called *Generalized Second Price (GSP)* payment rule, which charges the minimum threshold price below which the allocation changes. We prove that, even though our monotone allocation rule paired with GSP is not truthful, its Price of Anarchy (PoA) is bounded. Under standard no-overbidding assumptions, we prove bounds on the a pure and Bayes-Nash PoA. Finally, we experimentally test our algorithms on real-world data.

## 1   Introduction

Internet Ad Auctions, in addition to being influential in advancing auction theory and mechanism design, are a half-a-trillion dollar industry [CF21]. A significant advertising channel is sponsored search advertising: ads that are shown along with search results when you type a query in a search box. These ads traditionally were a few lines of text and a link, leading to the standard abstraction for ad auctions: multiple items for sale to a set of unit-demand bidders, where each bidder $i$ has a private value $v_i \cdot \alpha_{is}$ for the ad in position $s$, which has click-through rate $\alpha_{is}$. However, when using your favorite search engine, you might instead encounter sitelinks/extensions leading to parts of the advertisers' website, seller ratings indicating how other users rate this advertiser or offers for

36th Conference on Neural Information Processing Systems (NeurIPS 2022).

specific products. Advertisers can often explicitly opt in or out of showing different extensions with their ads. In fact, some extensions require the advertiser to provide additional assets, e.g. sitelinks, phone numbers, prices, promotion etc, and an ad cannot be shown unless this additional information is available. All these extensions/decorations change the amount of *space* the ad occupies, as well as affect the probability of a user clicking on the ad. The new and unexplored abstraction for modern ad auctions is now to select a set of ads that fit within the given total space.

In this paper, we study the problem of designing a *truthful* auction to determine the best set of ads that can be shown, with the goal of maximizing the social welfare. More formally, we consider the *Rich Ads* problem. In our model, each advertiser specifies a value per click $v_i$ and set of rich ads. Each ad has an associated probability of click $\alpha_{ij}$ and a space $w_{ij}$ that it would occupy if shown. The space and click probabilities are publicly known. Crucially, advertisers' private information is *not* single-dimensional. In addition to misreporting her value, there is another strategic dimension available: an advertiser can report only a subset of the set of ads available if the allocation under this report improves her utility. The open problem we address in this paper is whether there exist simple, approximately optimal and truthful mechanisms for Rich Ads.

**Results and Techniques.**   The classic Vickrey-Clarke-Groves (VCG) mechanism is truthful and maximizes welfare for our setting, but it is computationally intractable: maximizing welfare is NP-complete (even without truthfulness) since our problem generalizes the KNAPSACK problem. It is also well known that coupling an approximation algorithm for welfare with VCG payments does not result in a truthful mechanism. And, maximal-in-range mechanisms [NR01], that optimize social welfare over a restricted domain, even though are one way around such situations, have limited use, since the range of possible outcomes (allocations) has to be committed to before seeing the bidders preferences (i.e., needs to be independent of bidders reports). For single parameter problems, Myerson's lemma [Mye81] can be used to obtain a truthful mechanism, as long as the allocation rule is monotone. The Rich Ads problem is not a single parameter problem, so this approach does not immediately work. However, similar to the inter-dimensional (or "one-and-a-half" dimensional) regime [FGKK16, DHP20, DGS+20], we can extend Myerson's lemma to our domain. We show that an allocation rule that is monotone in the bid and the set of rich ads[1] can be paired with a payment rule to obtain a truthful mechanism.

Incentive issues aside, the Rich Ads problem is an extension of the KNAPSACK problem, called MULTI-CHOICE KNAPSACK: in addition to the knapsack constraint, we also have constraints that allow to allocate (at most) one rich ad per advertiser. As an algorithmic problem, this is well studied [SZ79, Law79]. The optimal fractional allocation can be derived using a simple *greedy* algorithm using the *incremental bang-per-buck* order. However, it turns out that the optimal (integral or otherwise) allocation, as well as other natural allocations, are not monotone. In fact, as we show, no deterministic (resp. randomized) monotone allocation rule can obtain more than half (resp. $11/12$ fraction) of the social welfare. In contrast, without the monotonicity constraint, there is an FPTAS for the MULTI-CHOICE KNAPSACK problem [Law79].

Our *main result* is providing an integral allocation rule that is monotone and obtains at least a third of the optimal (fractional) social welfare. Pairing with an appropriate payment function we get the following (informal) theorem.

**Informal Theorem.** *There exists a simple truthful mechanism, that can be computed in polynomial time, which obtains a $3$-approximation to the optimal social welfare.*

To obtain this result, we first find an allocation of space amongst the advertisers. In contrast to the optimal fractional algorithm described above which allocates greedily using the incremental bang-per-buck order, our algorithm allocates greedily using an *absolute bang-per-buck* order. Crucially, the *space* allocated to each advertiser in this way is monotone, even though the expected number of clicks (i.e. the utility) of the bang-per-buck algorithm itself is *not* monotone. By post-processing to utilize this space optimally for each advertiser, we obtain an integral allocation that is monotone in the expected number of clicks. We prove that this allocation gives a two approximation to the optimal fractional welfare, minus the largest value ad. Finally, by randomizing between this integral allocation (with probability $2/3$) and the largest value ad (with probability $1/3$), we get a 3-approximation

---

[1]Here, an allocation rule is defined to be monotone in the set of rich ads if the expected clicks allocated to an advertiser can only increase when the advertiser reports a superset of rich ads.

to the optimal social welfare. Since the overall allocation rule is monotone, we can pair it with an appropriate pricing to get a truthful mechanism.

We proceed to further explore the merits of our monotone allocation rule by pairing it with the *Generalized Second Price (GSP)* payment rule, which charges each advertiser the minimum threshold (on their bid) below which their allocation changes. The overall auction is not truthful. However, we can analyze its performance by bounding its social welfare in a worst-case equilibrium. In particular, we consider the full information pure Nash equilibrium, where bidders best-respond to a profile of competitors bids, as well as the incomplete information Bayes-Nash equilibrium, where the bidders best-respond to a distribution of valuation draws and bids for the competitors. The corresponding pure Price of Anarchy (PoA) and Bayes-Nash Price of Anarchy are the ratios of the optimal social welfare to the welfare of the worst equilibrium. In either setting, we make the standard no-overbidding assumption, where bidders do not bid more than their value. This assumption is required as without it the PoA of even the single-item second price auction (which is truthful) can be unbounded.[2]

**Informal Theorem.** *There exists a simple mechanism with a monotone allocation rule, paired with the GSP payment rule, which under the no-overbidding assumption guarantees a pure Price of Anarchy (resp. Bayes-Nash PoA) of at most $6$ (resp. $\frac{6}{1-1/e}$).*

We prove our PoA bounds by identifying a suitable deviation for each advertiser, and bounding the advertiser's utility in this deviation relative to the social welfare of the optimal integral allocation, our integral bang-per-buck allocation (in the equilibrium), and the largest value ad (in the equilibrium); as opposed to single-dimensional PoA bounds, the knapsack constraint in our setting introduces a number of technical obstacles we need to bypass. To prove a bound for the Bayes-Nash PoA, we combine techniques from our pure PoA bound with the standard smoothness framework [Rou15a]. In particular, the smoothness part of our argument is very similar to that of [CKK+15] for the Bayes-Nash PoA of the standard GSP position auction. Due to the specific form of the smoothness framework that we use, our bound also applies to mixed, correlated, coarse-correlated and Bayes-Nash with correlated valuations. Our PoA results can be found in Appendix E.

Finally, we provide an empirical evaluation of our mechanism on real world data from a large search engine. We compare performance of our mechanism with VCG and the fractional-optimal allocation that doesn't account for incentives. Our empirical results show that our allocation rule obtains at least $0.4$ fraction of the optimal in the worst-case. However, there are many instances where our allocation rule is almost as good as VCG. In fact, the average approximation factor of our allocation rule is $0.97$. Furthermore, our mechanisms are significantly faster than VCG, even with the Myersonian payment computation. We also empirically evaluate heuristic extensions of our algorithms when there is a bound on the total number of distinct rich ads shown.

**Related work.** Traditional sponsored search auctions have been studied extensively [AGM06, Var07, EOS07]. A number of recent works relax the traditional model of sponsored search auctions [CMSW20, Hum16] and introduce different versions of the "rich ads" problem [DSYZ10, CKSW17, GHLY19, HIK+18]; the specific model we study in this paper is new. [DSYZ10] are the first to formulate a rich ad problem where ads can occupy multiple slots. They analyze VCG and GSP variants for a special version of the rich ad problem where ads can be of only one of two possible sizes. They leave the problem addressed in this paper as an open problem for future work.

Much of the literature focuses on GSP-like rules (e.g., because the cost of switching from existing GSP to VCG can be high [VH14]). [CKSW17] consider the more general rich ad problem where there are constraints on number of ads shown and position effects in the click through rate. But their setting is still single-parameter — advertisers report a bid per click and cannot mis-report the set of rich ads. They provide a local search algorithm that runs within polynomial time and a generalized GSP like pricing to go with it. However, as opposed to our interest here, their auction is not truthful, nor do they give any approximation guarantees. [GHLY19] consider the optimization problem when the probability of click is submodular or subadditive in the size of the rich ad. They give an LP rounding based algorithm that provides a $4$ approximation for submodular and a $\Omega(\frac{\log m}{\log \log m})$ approximation for subadditive, with respect to the social welfare assuming truthful bidding. They however do not provide a truthful payment rule, or any PoA guarantees. These works also focus on a single-dimensional setting (where the advertiser is strategic about its bid but the set of ads is publicly

---

[2]Consider, for example, the equilibrium where all bidders bid $0$, except the lowest bidder, who bids infinity.

known). In contrast, we consider a multi-dimensional setting. Our simple and truthful mechanism also has a *monotone* allocation function, so we pair it with GSP as well.

The PoA of the GSP auction for text ads was studied in [CKKK11, LB10, LPL11]. Our PoA bounds use the smoothness framework introduced in [Rou15a], and later extended by [CKK$^+$15] to show PoA bounds for GSP (as well as in [Rou15b] and [ST13] for more general use).

## 2   Preliminaries

**Rich Ad Model.**   We introduce the following model for the rich ads auction problem. There is a set $\mathcal{N}$ of $n$ advertisers and a universe of rich ads $\mathcal{S}$. Each advertiser $i \in \mathcal{N}$ has a private value per click $v_i$ and a private set of rich ads $A_i \subseteq \mathcal{S}$.[3] We use $\mathbf{v} = (v_1, v_2, \ldots, v_n)$ to denote the vector of values per click and $\mathbf{A} = (A_1, A_2, \ldots, A_n)$ to denote the vector of sets of rich ads. For every advertiser $i \in \mathcal{N}$, each rich ad $j \in \mathcal{S}$ has a publicly known space $w_{ij}$ and a publicly known probability of click $\alpha_{ij}$.[4] We use $v_{ij}$ for the value of rich ad $j$ for advertiser $i$. If $j \notin A_i$, then the value of advertiser $i$ is $v_{ij} = 0$; otherwise, $v_{ij} = \alpha_{ij} v_i$. An advertiser can be allocated only one of the ads from the set $\mathcal{S}$. Finally, there is a total limit $W$ on the total space occupied by the ads. We assume without loss of generality that for each $i$ and each $j$, $w_{ij} \leq W$, as any ad that is larger than $W$ cannot be allocated integrally in space $W$. A (randomized or fractional) allocation $\mathbf{x} \in [0, 1]^{n \times |S|}$ indicates the probability $x_{ij}$ that ad $j$ is allocated to advertiser $i$. An allocation is feasible if each advertiser gets at most one ad, i.e. $\sum_{j \in \mathcal{S}} x_{ij} \leq 1$ for all $i \in \mathcal{N}$, and the total space used is at most $W$, i.e. $\sum_{i \in \mathcal{N}, j \in \mathcal{S}} x_{ij} w_{ij} \leq W$. An allocation is integral if $x_{ij} \in \{0, 1\}$, for all $i \in \mathcal{N}$ and $j \in \mathcal{S}$.

Our goal is to maximize social welfare. For an allocation $\mathbf{x} = Alg(\mathbf{v}, \mathbf{A})$, $SW(Alg(\mathbf{v}, \mathbf{A})) = \sum_{i,j} x_{i,j} v_{ij}$. We can write an integer program for the optimal allocation as follows, by introducing a binary variable $x_{ij} \in \{0, 1\}$ for the allocation of advertiser $i \in \mathcal{N}$ and rich ad $j \in \mathcal{S}$. The objective is to maximize welfare $\sum_{ij} x_{ij} v_{ij}$, subject to a Knapsack constraint $\sum_i \sum_j w_{ij} x_{ij} \leq W$, and feasibility, i.e. $\sum_j x_{ij} \leq 1$ for all $i \in \mathcal{N}$ (expressing that each advertiser can get only one ad).

**Mechanism Design Considerations.**   By standard revelation principle arguments, it suffices to focus on direct revelation mechanisms. Each advertiser $i \in \mathcal{N}$ reports a bid $b_i$ and a set of rich ads $S_i \subseteq \mathcal{S}$. Similarly to many works in the inter-dimensional regime, e.g. [MV09, DW17], we assume that $S_i \subseteq A_i$, that is, an advertiser cannot report that they want an ad they don't have. Let $\mathbf{b} = (b_1, b_2, \ldots, b_n)$ to denote the vector of bids and $\mathbf{S} = (S_1, S_2, \ldots, S_n)$ to denote the vector of sets of rich ads. We use $b_{ij} = b_i \cdot \alpha_{ij}$ if $j \in S_i$ and $b_{ij} = 0$ otherwise, and refer to the rich ad using a (reported value, space) tuple $(b_{ij}, w_{ij})$. A mechanism selects a set of ads to show, of total space at most $W$, and charges a payment to each advertiser. Let $x_{ij}(\mathbf{b}, \mathbf{S})$ be the probability that ad $j$ is allocated to advertiser $i$, and $p_i(\mathbf{b}, \mathbf{S})$ denote the expected payment of advertiser $i$. Let $\mathbf{x}_i(\mathbf{b}, \mathbf{S})$ be the allocation vector of advertiser $i$. We assume that for any valid allocation rule for $j \notin S_i$ $x_{ij}(\mathbf{b}, \mathbf{S}) = 0$. We slightly overload notation, and use $x_i(\mathbf{b}, \mathbf{S})$ to denote the expected number of clicks the advertiser will get; that is, $x_i(\mathbf{b}, \mathbf{S}) = \sum_{j \in S_i} x_{ij}(\mathbf{b}, \mathbf{S}) \alpha_{ij}$. If required we refer to the cost per-click $cpc_i(\mathbf{b}, \mathbf{S}) = p_i(\mathbf{b}, \mathbf{S}) / x_i(\mathbf{b}, \mathbf{S})$

Advertisers have quasi-linear utilities. An advertiser with value $v_i$ and set $A_i$ has utility $v_i x_i(\mathbf{b}, \mathbf{S}) - p_i(\mathbf{b}, \mathbf{S})$, when reports are according to $\mathbf{b}$ and $\mathbf{S}$. Let $u_i(v_i, A_i \to b_i, S_i; \mathbf{b}_{-i}, \mathbf{S}_{-i})$ be the utility of advertiser $i$ when her true value and set of ads are $v_i, A_i$, but reports $b_i, S_i$, and everyone else reports according to $\mathbf{b}_{-i}, \mathbf{S}_{-i}$. For ease of notation we often drop $\mathbf{b}_{-i}, \mathbf{S}_{-i}$ when it's clear from the context. When the profile of true types is fixed, we drop $(v_i, A_i)$ and use the notation $u_i(b_i, S_i, \mathbf{b}_{-i}, \mathbf{S}_{-i})$.

A mechanism is *truthful* if no advertiser has an incentive to lie, i.e. for any all $\mathbf{b}_{-i}, \mathbf{S}_{-i}$, $u_i(v_i, A_i \to v_i, A_i; \mathbf{b}_{-i}, \mathbf{S}_{-i}) \geq u_i(v_i, A_i \to b_i, S_i; \mathbf{b}_{-i}, \mathbf{S}_{-i})$, for all $v_i, A_i, b_i, S_i$, A mechanism is individually rational if in all of its outcomes, all agents have non-negative utility.

We are interested in auctions that are computationally tractable, truthful, individually rational, with the goal of maximizing the social welfare $SW(\mathbf{x}(\mathbf{b}, \mathbf{S})) = \sum_i v_i x_i(\mathbf{b}, \mathbf{S})$. Even ignoring truthfulness

---

[3]We expect the rich ads to be tailored to an advertiser, so we assume that $A_i \cap A_{i'} = \emptyset$, for all $i, i' \in \mathcal{N}$.

[4]Note that this safe to assume. The space consumed by a rich ad is evident when the rich ad is provided. The probability of click can be predicted by the platform (e.g. using machine learning models).

and individual rationality, the computational constraints rule out achieving the optimal social welfare; therefore, we seek approximately optimal mechanisms.

**Definition 1.** *A truthful mechanism $\mathcal{M}$ obtains an $\alpha$ factor approximation to the social welfare if $SW(\mathbf{x}_{\mathcal{M}}(\mathbf{v}, \mathbf{A})) \geq SW(\mathbf{x}_{OPT}(\mathbf{v}, \mathbf{A}))/\alpha$.*

Formal definitions of the Generalized Second Price and the Price of Anarchy are deferred to Appendix A (since all the instances where they are used are also in the appendix).

**Optimal fractional allocations.**    The algorithmic problem is a special case of a well-known variation of the KNAPSACK problem, called MULTI-CHOICE KNAPSACK [SZ79]. The integer program for MULTI-CHOICE KNAPSACK is the same as the integer program above, except that the inequality constraint $\sum_j x_{ij} \leq 1$ is replaced with an equality. Our problem is easily reduced to the MULTI-CHOICE KNAPSACK problem by introducing a *null* ad with $(\alpha_{i0}, w_{i0}) = (0, 0)$.[SZ79] provide a characterization of the optimal fractional solution of MULTI-CHOICE KNAPSACK and provide a fast algorithm to compute the fractional optimal solution. As is colloquial in the KNAPSACK literature, we refer to the ratio $\frac{b_{ik}}{w_{ik}}$ as *Bang-per-Buck* and the ratio $\frac{b_{ij}-b_{ik}}{w_{ij}-w_{ik}}$ with $w_{ij} > w_{ik}$ as *Incremental Bang-per-Buck*. [SZ79] show that allocating ads in the incremental bang-per-buck order gives the optimal fractional solution. We state this standard algorithm in Appendix A. We refer to the solution constructed by this algorithm as OPT and use it as a benchmark in our approximation guarantees. We note a few more properties of OPT.

**Fact 1** ([SZ79])**.** *In the optimal fractional allocation constructed by the algorithm, all advertisers except one have a rich-ad allocated integrally. Also for any advertiser $i$ allocated space $W_i^*$ in OPT, the allocation maximizes the value that advertiser $i$ can obtain in that space.*

This fact also implies a 2-approximate integral allocation as follows. Construct an optimal fractional solution using the incremental bang-per-buck order. Let $i'$ denote the advertiser that is allocated last: select the larger of the optimal fractional solution without $i'$ and the highest value ad of $i'$.

## 3   Monotonicity and Lower Bounds

**Monotonicity implies truthfulness.**    In single-parameter domains, Myerson's lemma provides a handy tool for constructing truthful mechanisms. One has to only construct a monotone allocation rule, and then the lemma provides a complementary payment rule such that the overall mechanism is truthful. We extend this approach to our particular multi-parameter domain. If the set of ads is fixed for each advertiser, then monotonicity in bid and Myerson-like payments imply truthfulness. We give constraints between the allocation rules for different sets of ads and show that they imply truthfulness everywhere using a local-to-global argument. We begin by defining monotonicity in our setting. An allocation rule is said to be monotone if it is monotone in each dimension of the buyer's preferences.

**Definition 2.** *An allocation rule $x(\mathbf{b}, \mathbf{S})$ is monotone in $b_i, S_i$ for each $i$, if (1) For all $\mathbf{b}_{-i}, \mathbf{S}_{-i}, S_i, b'_i \geq b_i$; we have $x_i(b'_i, S_i, \mathbf{b}_{-i}, \mathbf{S}_{-i}) \geq x_i(b_i, S_i, \mathbf{b}_{-i}, \mathbf{S}_{-i})$, and (2) For all $\mathbf{b}_{-i}, \mathbf{S}_{-i}, b_i, S'_i \supseteq S_i$; we have $x_i(b_i, S'_i, \mathbf{b}_{-i}, \mathbf{S}_{-i}) \geq x_i(b_i, S_i, \mathbf{b}_{-i}, \mathbf{S}_{-i})$.*

As the following example shows, the optimal allocation rule is *not* monotone. The example also shows that monotonicity is not necessary for truthfulness (since VCG is truthful and optimal).

**Example 1.** *Consider two advertisers with two rich ads each. Both have value 1 and the rich ads have (value, size) $= (1, 1), (1 + \epsilon, 2)$. The space available is 3. In this case, the optimal integer solution chooses the smaller ad from one advertiser and the larger ad from other. However, if one of them removes their smaller option, they get the larger option deterministically.*

The next lemma (proof in Appendix B) shows that monotonicity is sufficient for truthfulness.

**Lemma 1.** *If a valid allocation rule $\mathbf{x}(\mathbf{b}, \mathbf{S})$ is monotone in $b_i, S_i$ for each $i \in \mathcal{N}$, then charging payment $p_i(\mathbf{b}, \mathbf{S}) = b_i x_i(\mathbf{b}, \mathbf{S}) - \int_0^{b_i} x_i(b, \mathbf{b}_{-i}, \mathbf{S})db$ results in a truthful auction.*

**Lower bounds.**    Next, we illustrate the challenge in coming up with allocation rules which are monotone in the set of rich ads. We also prove that monotonicity rules out approximation ratios strictly better than 2 for deterministic mechanisms (and $12/11$ for randomized mechanisms).

As we've seen in Example 1, the algorithm that finds the optimal integer allocation is not monotone. The following example shows that simple algorithms such as selecting ads in the incremental bang-per-buck order, or the 2 approximation algorithm presented in Section 2 are not monotone either. Recall that bang-per-buck = $b_{ij}/w_{ij}$ and incremental bang-per-buck = $\frac{b_{ij}-b_{ik}}{w_{ij}-w_{ik}}$ with $w_{ij} > w_{ik}$.

**Example 2.** *We have two advertisers A and B. A has two rich ads with (value, size) as (2,1),(3.5,3). B has one ad with (value, size)=(3,3).*

*(i) Suppose $W = 4$. The incremental bang-per-buck algorithm picks $A : (2, 1)$, followed by $B : (3, 3)$. The rich ad $A : (3.5, 3)$ does not get picked over $B : (3, 3)$, despite having higher bang-per-buck, since it has smaller incremental bang-per-buck (of $0.75$). On the other hand if A removes the rich ad $(2, 1)$, the algorithm picks $A : (3.5, 3)$ at the very beginning, and A obtains a higher value.*

*(ii) Suppose $W = 3.5$, the optimal fraction solution is $A : (2, 1)$ with a weight of $1.0$ and $B : (3, 3)$ with weight $2.5/3$. The 2-approximation algorithm of Section 2 compares allocating just $A : (2, 1)$ or just $B : (3, 3)$, and chooses B. But if A removed $(2, 1)$, then the optimal fraction solution is $A : (3.5, 3)$ with a weight of $1.0$ and $B : (3, 3)$ with weight $0.5/3$. The 2-approximation algorithm compares $A : (3.5, 3)$ and $B : (3, 3)$, and chooses A because it has higher value. Thus A gets a higher value by removing $(2, 1)$.[5]*

The next theorem gives lower bounds on the approximation factor a monotone algorithm can achieve.

**Theorem 1.** *No monotone and deterministic (resp. randomized) algorithm has an approximation ratio better than $2 - \varepsilon$ (resp. $12/11 - \varepsilon$) for any $\varepsilon > 0$.*

*Proof.* There are two advertisers. Each advertiser has two rich ads: $(1, 1), (1 + \varepsilon, 2)$. The total space is 3. The optimal solution has value $(2 + \epsilon)$. To obtain an approximation better than $2 - \epsilon$, we must choose a small ad for at least one of the advertisers. Since the algorithm is monotone, when that advertiser does not provide the small ad, the algorithm cannot give them the larger ad. Thus when the advertiser does not provide the small ad, the algorithm must give this advertiser nothing, resulting in welfare at most $(1 + \varepsilon)$. See Appendix B for the proof for randomized algorithms. $\square$

## 4   A Simple Monotone $3$-Approximation

In this section we give our main result: a monotone algorithm that obtains a 3 approximation to the optimal social welfare. First, we give a fractional algorithm for allocating *space* to each advertiser $i$. Second, we show that optimally (and integrally) using the space given to each advertiser $i$ gives a monotone allocation. Finally, we show that randomizing between the former algorithm and simply allocating the max value ad is a monotone rule that obtains a 3 approximation to $OPT$. Omitted proofs (and definitions) can be found in Appendix C. We show that the truthful payment function matching our allocation rule can be computed in polynomial time in Appendix C.3.

**Monotone space allocation algorithm**   We start by giving a monotone algorithm for allocating space to each advertiser. This total space allocated is monotone in $b_i$ and $S_i$, for all $i \in \mathcal{N}$. Our algorithm also provides an allocation of rich-ads to that space, but this allocation by itself may not be monotone, and may not provide a good approximation. Our algorithm, which we call $ALG_B$, works as follows (see Appendix C for a formal description). First, we order the ads in the bang-per-buck order. We iteratively choose the next ad in this order; let $i$ be the corresponding advertiser, and $j$ be the rich ad. We replace the previous ad of $i$ with $j$, if this choice results in more space allocated to $i$. If there is not enough space we fractionally allocate $j$ and terminate.

The following observation shows that any ads removed for not being-selected will not be used by the fractional-optimal solution as well. See Appendix A for the precise definition of dominated.

**Observation 1.** *Let $j' \in E_i$ be some ad removed from $E_i$ in "step 2" of $ALG_B$ for having space at most $w_{ij}$. Then either $j' = j$ or $j'$ is a "dominated" ad.*

Next, we prove that $ALG_B$ allocates space that is monotone in $b_i$ and $S_i$, for all $i \in \mathcal{N}$.

---

[5]Even though we only seek integer monotone algorithms, this example shows that even the fractional allocation is not monotone: when A provides all ads its value is 2, but when A removes $(2, 1)$, its value is 3.5.

**Theorem 2.** *Let $\mathbf{x}(ALG_B)$ denote the allocation of $ALG_B$. Then, for all $i \in \mathcal{N}$, $b_i \leq b_i'$: $\sum_j w_{ij} x_{ij}(ALG_B(\mathbf{b}, \mathbf{S})) \leq \sum_j w_{ij} x_{ij}(ALG_B(b_i', \mathbf{b}_{-i}, \mathbf{S}))$. Also, for all $i$ and $S_i \subseteq S_i'$, the space $W_i$ is monotone in $S_i$: $\sum_j w_{ij} x_{ij}(ALG_B(\mathbf{b}, \mathbf{S})) \leq \sum_j w_{ij} x_{ij}(ALG_B(\mathbf{b}, S_i, \mathbf{S}_{-i}))$.*

$ALG_B$ is inefficient since the bang-per-buck allocation can change the relative order in which the advertisers are assigned space, generating sub-optimal outcomes.

**Example 3.** *Let $M$ be a large integer. Let $W = M - 1$ and consider two advertiser $A$ and $B$. $A$ has two rich ads with (value, size) as $(1, 1), (1 + \varepsilon, M - 1)$. $B$ has one rich ad with (value, size) $= (\frac{M-1}{M}, M - 1)$. Fractional OPT selects $A : (1, 1)$ fully and $B : (\frac{M-1}{M}, M - 1)$ fractionally with weight $(M - 2)/(M - 1)$ and obtains a social welfare $1 + \frac{M-2}{M}$. The bang-per-buck allocation in $ALG_B$ selects $A : (1 + \varepsilon, M)$ fully and obtains social welfare $(1 + \varepsilon)$.*

While the space allocated is monotone in the bid and rich ads, the allocation itself may not be monotone, since $ALG_B$ may allocate an advertiser a larger ad with lower value than another option; we provide an example in Appendix D.

**Integral Monotone Allocation.** The allocation generated by $ALG_B$ can be fractional for one advertiser, and can be sub-optimal (but integer) for some of the other advertisers. In the following algorithm, we post-process to find the best *single ad* that fits in $W_i$.

**Algorithm 1** ($ALG_I$). *First, run $ALG_B$. Let $W_i$ be the space allotted to advertiser $i$. Second, post-process to allocate the ad $j$ with maximum value that fits in $W_i$, i.e. $j \in \operatorname{argmax}_{w_{ij} \leq W_i} b_{ij}$. Any remaining space is left unallocated.*

Observe that $ALG_I$ is monotone, since (1) the space allocated by $ALG_B$ is monotone, and (2) if the space allocated by $ALG_B$ is larger, then the post-processing that allocates the highest value ad that fits in this space will also result in same or larger value. However, $ALG_I$ alone might be an arbitrarily bad approximation: we provide an example in Appendix D.

**Main result.** Our main result is the following theorem.

**Theorem 3.** *The randomized algorithm that runs $ALG_I$ with probability $2/3$, and otherwise allocates the maximum valued ad, is monotone in $b_i$ and $S_i$, obtains a 3-approximation to the social welfare, and this approximation factor is tight.*

*Proof.* Let $b_{max}$ denote the value of the maximum valued ad. We will first show that $2SW(\mathbf{x}(ALG_I)) + b_{max} \geq SW(\mathbf{x}_{OPT})$. Let $\operatorname{Val}(\mathbf{x}, A, \vec{s}) = \sum_{i \in A} (\sum_{j \in \mathcal{S}} b_{ij} \cdot x_{ij}) \cdot \left( \frac{s_i}{\sum_{j \in \mathcal{S}} w_{ij} x_{ij}} \right)$ be the fraction of the social welfare of allocation $\mathbf{x}$, $SW(\mathbf{x})$, contributed by a subset of advertisers $A$ for space $\vec{s}$. Let $\mathbf{x}^* = \mathbf{x}_{OPT(\mathbf{b}, \mathbf{S})}$ denote an optimal fractional allocation. Let $W_i = W_i(ALG_B(\mathbf{b}, \mathbf{S}))$ and $W_i^* = W_i(OPT(\mathbf{b}, \mathbf{S}))$ denote the total space allocated to advertiser $i$ in $ALG_B$ and the optimal allocation $\mathbf{x}^*$, respectively. The space allocated in $ALG_I$ is exactly the same as $W_i$. Recall Fact 1, that there is an optimal fractional allocation where at most one advertiser is allocated fractionally. Let $i'$ be the advertiser whose allocation in $\mathbf{x}^*$ is fractional. There is also at most one advertiser in $ALG_B$ whose allocation is fractional: the advertiser corresponding to the very last ad that is included; let $i''$ be this advertiser. We start by giving a series of technical claims. First, we bound the part of $SW(\mathbf{x}(OPT))$ contributed by advertisers who are allocated more space in $ALG_B$ than in $OPT$. Let $\mathcal{I}$ denote the set of advertisers $i$ with $W_i \geq W_i^*$.

**Claim 1.** $\operatorname{Val}(\mathbf{x}^*, \mathcal{I} \setminus \{i'\}, \vec{W}^*) \leq \operatorname{Val}(\mathbf{x}(ALG_I), \mathcal{I} \setminus \{i'\}, \vec{W})$.

Let $\mathcal{K} = \mathcal{N} \setminus \mathcal{I}$ be the set of advertisers $k$ with $W_k < W_k^*$. If $\mathcal{K} = \emptyset$ then $W_i = W_i^*$ for all $i$. We note that $b_i \cdot x_i(ALG_I) = b_i \cdot \mathbf{x}_i^*$ for all $i \neq i'$ by Claim 1. Also $b_{max} \geq b_i \cdot \mathbf{x}_{i'}^*$. Thus we get $SW(\mathbf{x}(ALG_I)) + b_{max} \geq SW(OPT)$. So for the rest of the proof we assume $\mathcal{K} \neq \emptyset$. We bound the portions of $SW(\mathbf{x}_{OPT})$ contributed by $k \in \mathcal{K}$ using the following claim.

**Claim 2.** *For all $k \in \mathcal{K} \setminus \{i'\}$ and $i \in \mathcal{N}$, we have $\frac{b_k \cdot \mathbf{x}_k^*}{W_k^*} \leq \frac{b_i \cdot x_i(ALG_B)}{W_i}$.*

That is, advertisers that are allocated less space in $ALG_I$ than in $\mathbf{x}^*$, must have lower bang-per-buck as otherwise their rich ad would be considered by $ALG_I$ and they will be allocated more space.

Finally we bound the contribution of $i'$, that is the ad allocated fractionally in $OPT$ as follows:

**Claim 3.** *Let $(b_s, s), (b_\ell, \ell)$ with $s < l$ be the ads used in $\mathbf{x}_{i'}^*$, the optimal fractional allocation for $i'$. It holds that: (i) $\frac{b_\ell - b_s}{\ell - s} \leq \frac{b_i \cdot \mathbf{x}_i(ALG_B)}{W_i}$ for all $i \in \mathcal{N}$, (ii) if $s > W_{i'}$, then $\frac{b_s}{s} \leq \frac{b_i \cdot \mathbf{x}_i(ALG_B)}{W_i}$ for all $i \in \mathcal{N}$, and (iii) if $s \leq W_{i'}$ then $b_s \leq b_{i'} \cdot \mathbf{x}_{i'}(ALG_I)$.*

The proof of this claim is more involved. Intuitively we can bound the smaller of $i'$'s allocated ads (if it is small enough) with the value $i'$ obtains in $ALG_I$ and the larger ad since it is allocated last has the lowest incremental bang-per-buck than any other advertiser's ad and hence the incremental bang-per-buck is also lower than the other ad's actual bang-per-buck.

We put all the claims together to bound $\text{Val}(\mathbf{x}^*, \mathcal{K} \cup \{i'\}, \vec{W}^*)$. If $s > W_{i'}$, then by Claims 2 and 3 we see that the bang-per-buck of $k \in \mathcal{K} \cup \{i'\}$ is less $b_i \cdot \mathbf{x}_i(ALG_B)/W_i$ for all $i$. Then the value for $\mathcal{K} \cup \{i'\}$ in OPT is less than the contribution of advertisers using the *same total space* in $ALG_B$ .

$$\text{Val}(\mathbf{x}^*, \mathcal{K} \cup \{i'\}, \vec{W}^*) \leq \text{Val}(\mathbf{x}^*, \mathcal{K} \setminus \{i'\}, \vec{W}^*) + b_{i'} \cdot \mathbf{x}_{i'}^* \leq \text{Val}(\mathbf{x}(ALG_B), \mathcal{N}, \vec{W}) \quad (1)$$

If $s \leq W_{i'}$, then by Claims 2 and 3 we get that $b_s < b_{i'} \cdot \mathbf{x}_{i'}(ALG_I)$ and the bang-per-buck of the ads in $ALG_B$ is higher than that of $K \setminus \{i'\}$ and $\frac{b_\ell - b_s}{\ell - s}$. Thus we have,

$$\text{Val}(\mathbf{x}^*, \mathcal{K} \cup \{i'\}, \vec{W}^*) \leq \text{Val}(\mathbf{x}^*, \mathcal{K} \setminus \{i'\}, \vec{W}^*) + b_s + (W_{i'}^* - s)\frac{b_\ell - b_s}{\ell - s}$$

$$\leq \text{Val}(\mathbf{x}(ALG_B), \mathcal{N}, \vec{W}) + b_{i'} \cdot \mathbf{x}_{i'}(ALG_I) \quad (2)$$

Hence by putting both $\mathcal{I}$ and $\mathcal{K}$ together we get,

$$SW(\mathbf{x}_{OPT}) = \text{Val}(\mathbf{x}^*, \mathcal{I} \setminus \{i'\}, \vec{W}^*) + \text{Val}(\mathbf{x}^*, \mathcal{K} \cup \{i'\}, \vec{W}^*)$$

$$\leq \text{Val}(\mathbf{x}(ALG_I), \mathcal{I} \setminus \{i'\}, \vec{W}) + \text{Val}(\mathbf{x}(ALG_B), \mathcal{N}, \vec{W}) + b_{i'} \cdot \mathbf{x}_{i'}(ALG_I)$$

$$\leq \text{Val}(\mathbf{x}(ALG_I), \mathcal{N}, \vec{W}) + \text{Val}(\mathbf{x}(ALG_B), \mathcal{N}, \vec{W})$$

$$\leq 2\text{Val}(\mathbf{x}(ALG_I), \mathcal{N}, \vec{W}) + b_{max}$$

where the first equality is the definition of $SW(\mathbf{x}_{OPT})$, and the second inequality puts together Claim 1 and Equations (1), and (2). The third inequality holds because $\mathcal{I} \cup \{i'\} \subseteq \mathcal{N}$. The final inequality holds because $b_i \cdot \mathbf{x}_i(ALG_I) \geq b_i \cdot \mathbf{x}_i(ALG_B)$ for all $i \neq i''$, and $b_{i''} \cdot \mathbf{x}_{i''}(ALG_B) \leq b_{max}$.

Thus we have that $SW(\mathbf{x}_{OPT}) \leq 2\text{Val}(\mathbf{x}(ALG_I), \mathcal{N}, \vec{W}) + b_{max}$. Therefore, running $ALG_I$ with probability $2/3$ and allocating $b_{max}$ with probability $1/3$ is a 3-approximation to $SW(\mathbf{x}_{OPT})$. Since our algorithm is randomizing between two monotone rules, it is monotone as well. We give an instance which shows that the approximation of the algorithm is at least 3 in Appendix C.  □

## 5 Experiments

In this section we present some empirical results for our truthful mechanisms. The allocation rules we use for our theoretical results can be extended to obtain higher value. First we extend $ALG_I$ to skip past a high bang-per-buck ad that does not fit in the remaining space. More precisely, recall that $ALG_I$ calls $ALG_B$ to get the space allocation. $ALG_B$ stops when the ad being considered cannot be fit fully in the remaining space. The advertiser corresponding to this ad still gets allocated the remaining space, which is filled with the highest-value ad that fits in post-processing. In our modified version, we update $ALG_B$ to skip past this large ad. This is equivalent to dropping step (3) of $ALG_B$ (i.e., we keep going until we run out of ads), and running the rest of $ALG_I$ as it is. We call this modified algorithm *GreedyByBangPerBuck*. For our theoretical result we also select the maximum value ad with probability $1/3$. In practice, this can be very inefficient. For our empirical evaluation, we extend this to continues to allocate as long as space is remaining. Similar to the GreedyByBangPerBuck, the algorithm skips past a high value but large ad that cannot fit, and continues allocating until the space or the set of rich ads runs out. We call this algorithm *GreedyByValue*. It is worth noting that these extensions do not improve the worst-case approximation ratio of Theorem 3: we include a brief proof in Appendix F. We implement our proposed randomized algorithm by flipping a coin with probability $2/3$ for each query and selecting the result of GreedyByBangPerBuck algorithm if it is heads and GreedyByValue otherwise. We call this *RandomizedGreedy*. As a baseline, we

| Algorithm | ApproxECPM | time-msec |
|---|---|---|
| GreedyByBangPerBuck | 0.9493 | 0.7363 |
| GreedyByValue | 0.9196 | 0.3242 |
| RandomizedGreedy | $0.9393 \pm 0.0001$ | $0.5983 \pm 0.0007$ |
| IntOPT | 1.0 | 6.9988 |

Table 1: Average performance of the algorithms compared to IntOPT. We report average approximation of eCPM relative to IntOPT and average running time in miliseconds. We report confidence intervals for the randomized algorithm by noting the average performance over all queries over 100 runs.

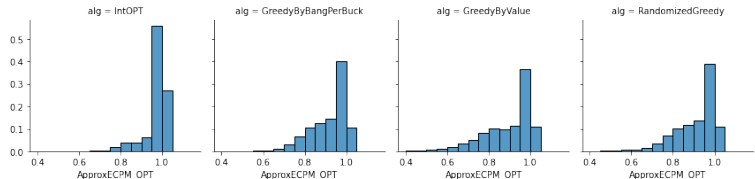

Figure 1: Histogram of approximation factor for IntOPT, GreedyByValue, GreedyByBangPerBuck and RandomizedGreedy compared to Fractional Optimal.

have implemented *IntOPT*, that uses brute-force-search to evaluate all possible allocations and compute the optimal integer allocation. This allocation paired with the VCG payment rule (which is computed by computing the integer OPT with advertiser $i$ removed, and subtracting from that the allocation of all advertisers other than $i$ in the optimal allocation) gives the VCG mechanism. Finally, we implement the incremental bang-per-buck order algorithm as *IncrementalBPB* to compute the fractionally optimal allocation.

We evaluated our algorithms from real world data obtained from a large search advertising company. The data consists of a sample of approximately 11000 queries, selected to have at least 6 advertisers each. All the space values for the rich-ads are integer. We use 500 as the space limit as that is larger than the space of any individual rich ad. Table 1 compares the average performance of these algorithms. Our algorithms are comparable to IntOPT, on average, but require a lot less time to run.

We first compare the approximation obtained by various algorithms to the fractional-optimal. In Figure 1, we see that GreedyByBangPerBuck and IntOPT obtain at least a $0.55$ fraction of the fractional opt, while the approximation factor for GreedyByValue can be as low as $0.4$. There is also a substantial amount of mass in the $1.0$ bucket where integer-opt and fractional-opt coincide and the greedy algorithms also sometimes achieve that. Next we compare the approximation obtained by various algorithms to IntOPT.

In Figure 2 we see the approximation obtained by various algorithms compared to the IntOPT allocation. There are more queries where we obtain the optimal approximation, but the worst-case is still $0.6$ for GreedyByBangPerBuck and $0.4$ for GreedyByValue. For additional insight, we plot a heatmap to correlate the approximation factor obtained by GreedyByBangPerBuck and GreedyByValue with IntOPT as the baseline. In Figure 2, we compare the approximation factor of

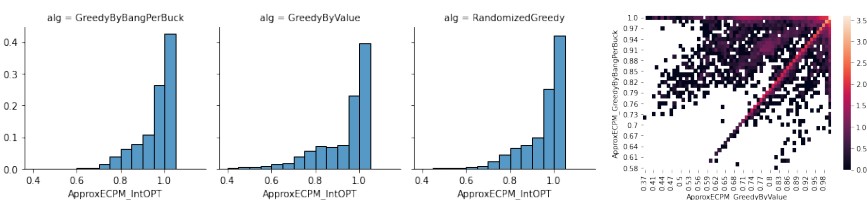

Figure 2: Histogram of approximation factor for GreedyByValue, GreedyByBangPerBuck and randomized Greedy compared to IntOPT. Height of each bucket represents the fraction of queries in the bucket. Last figure shows correlation of approximation factor relative to IntOPT for GreedyByValue, GreedyByBangPerBuck. Color-scale in the heat-map by log(number of queries) in bucket.

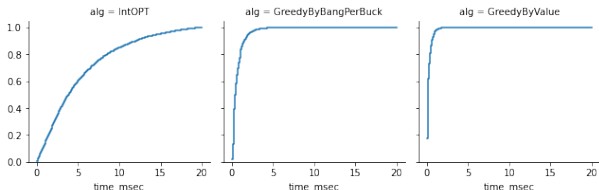

Figure 3: CDF of running time in milliseconds for for GreedyByValue, GreedyByBangPerBuck and IntOPT algorithms.

GreedyByValue and GreedyByBangPerBuck. Blank spaces in this plot correspond to not having any queries with a particular combination of approximation factors. We note that a lot of the queries have the same approximation factor for GreedyByValue and GreedyByBangPerBuck — indicating that RandomizedGreedy won't make mistakes. But GreedyByBangPerBuck more often has better approximation factor than GreedyByValue, so sticking to GreedyByBangPerBuck as the only heuristic might perform better. Finally, in Figure 3 we compare the clock-time of our allocation rules with that of the IntOPT allocation rule.

These allocation rules are monotone, so they can be paired with Myerson's payment rule as implied by Lemma 1. We evaluate the time required to compute the payments and relative revenue compared to VCG in Appendix G (Table 3). Since the focus of our paper is welfare maximization we do not consider reserve prices. A possible direction for future work is to study the trade-offs between welfare and revenue of our mechanism under different reserve prices. Additionally, in Appendix G we evaluate our algorithms with an added cardinality constraints and show that we obtain reasonable approximations to the optimal allocation.

## Acknowledgements

The work of Divyarthi Mohan was partially supported by the European Research Council (ERC) under the European Union's Horizon 2020 research and innovation program (grant agreement no. 866132) and by the Israel Science Foundation (grant no. 317/17). Alexandros Psomas is supported in part by an NSF CAREER award CCF-2144208, a Google Research Scholar Award, and by the Algorand Centres of Excellence program managed by Algorand Foundation. Any opinions, findings, and conclusions or recommendations expressed in this material are those of the author(s) and do not necessarily reflect the views of Algorand Foundation. Part of this work was done when Mohan and Psomas were employed at Google Research.

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
