# A  Missing Preliminaries

## A.1  Fractional Optimal and the Incremental-bang-per-buck Algorithm

[SZ79] introduce the notion of *Dominated* and *LP dominated* ads and show that they are not used in the fractional optimal solution.

**Definition 3** (Dominated/LP-dominated [SZ79])**.** *For an advertiser $i$, the rich ads of the advertiser can be dominated in two ways.*

- *Dominated: If two rich ads satisfy $w_{ij} \leq w_{ik}$ and $b_{ij} \geq b_{ik}$ then $k$ is dominated by $j$.*

- *LP Dominated: If three rich ads $j, k, l$ with $w_{ij} < w_{ik} < w_{il}$ and $b_{ij} < b_{ik} < b_{il}$, satisfy $\frac{b_{il} - b_{ik}}{w_{il} - w_{ik}} \geq \frac{b_{ik} - b_{ij}}{w_{ik} - w_{ij}}$, then $k$ is LP-dominated by $j$ and $l$.*

Moroever, [SZ79] showed that the fractional optimal solution can be obtained through the following *incremental-bang-per-buck* algorithm.

**Algorithm 2.**

- *Eliminate all the Dominated and LP-dominated rich ads for each advertiser.*

- *(Compute incremental-bang-per-buck) For each advertiser, allocate the null ad. Sort the remaining rich ads by space (label them $i1, \ldots ik, \ldots$). Construct a vector of scores $\frac{b_{ik} - b_{ik-1}}{w_{ik} - w_{ik-1}}$ for these.*

- *(Allocate in incremental bang-per-buck order) While space remains: select the rich ad $(i, k)$ with highest score among remaining rich ads. If the remaining space is at least as much as the incremental space required ($w_{ik} - w_{i(k-1)}$), this new rich ad is allocated to its advertiser and it fully replaces previously allocated rich ad for the advertiser. Otherwise, allocate the advertiser fractionally in the remaining space. This fractional allocation puts a weight $x$ on the previously allocated rich ad of the advertiser and $(1-x)$ on the newly selected rich-ad such that the fractional space equals remaining space plus previously allocated space of the advertiser.[6]*

We prove some lemmas about the optimal fractional solution. The following lemma provides a simple characterization of the advertisers' welfare depending on whether the optimal solution uses one or two rich ads fractionally.

**Lemma 2.** *Let $W_i^*$ be the space allocated to advertiser $i$ with a non-null allocation in an optimal solution to the* MULTI-CHOICE KNAPSACK *problem.*

*There are two cases.*

1. *The optimal allocation uses a single non-null rich ad with (value, size) $(b_l, l)$ ($l = W_i^*$) in space $W_i^*$. The advertiser is allocated integrally and its value is $b_l$.*

2. *The optimal allocation uses two rich ads with (value, size) $(b_s, s)$, $(b_l, l)$, with $s < W_i^* < l$ and $(b_l, l)$ not null, in space $W_i^*$. We have that $b_l > b_s$ and the advertiser's value (i.e. their contribution to the social welfare) is $b_s + \frac{b_l - b_s}{l - s}(W_i^* - s)$. If $(b_s, s)$ is not null, $\frac{b_s}{s} \geq \frac{b_l}{l} \geq \frac{b_l - b_s}{l - s}$.*

*Proof.* Recall that the main purpose of the null ad is to help make the fractional allocation of advertiser $i$ exactly equal to one. We can reason about the optimal allocation of advertiser $i$ in space $W_i$ without the null ad and bringing the null ad if $i$'s allocation is less than one. Note that the null ad does not change the value or space occupied by advertiser $i$. The optimization problem for a single

---

[6]For more details refer to Lemma 2 in the appendix.

advertiser (without the null ad) is as follows. Let $S$ be the set of rich ads for advertiser $i$.

$$\max \sum_{k \in S} x_k b_k$$

$$s.t. \sum_{k \in S} x_k w_k \leq W_i^*$$

$$\sum_{k \in S} x_k \leq 1$$

$$x_k \geq 0 \qquad \forall k \in S$$

This LP has $|S|$ variables and $|S| + 2$ constraints, so there exists an optimal solution with at most 2 non-zero variables.

Suppose there is only one non-zero variable. The optimal fractional solution is made of a single ad $(b_l, l)$ with $s \geq W_i^*$. Suppose the advertiser is allocated $x \leq 1$. Then since $lx \leq W_i^*$, we have that $x \leq \frac{W_i^*}{l}$. Since the optimal fractional solution maximizes the advertiser $i$'s value $b_l x$ in that space (as noted in Fact 1), we have that $x = \frac{W_i^*}{l}$ and the advertisers value is $b_l \frac{W_i^*}{l} = \frac{b_l}{l} W_i^*$. When $l = W_i^*$, $x = 1$ and the advertiser's allocation is integral. Otherwise, $x < 1$, we can set $(b_s, s) = (0, 0)$ with $x_s = 1 - x$ and the advertiser's value is still $\frac{b_l}{l} W_i^* = \frac{b_l - 0}{l - 0} W_i^* + 0$.

Suppose the optimal allocation uses two (non-null) rich ads $(b_s, s)$, $(b_\ell, \ell)$. Then both the knapsack and unit demand constraints must be tight. That is $x_s s + x_\ell \ell = 1$ and $x_s s + x_\ell \ell = W_i^*$. The solution to this system is to have $x_s = \frac{l - W_i^*}{l - s}$ and $x_l = \frac{W_i^* - s}{l - s}$. Both $x_s$ and $x_l$ are fractional if $s < W_i^* < l$. And the advertiser's value is

$$b_s \frac{l - W_i^*}{l - s} + b_l \frac{W_i^* - s}{l - s}$$

$$= b_s + (b_l - b_s) \frac{W_i^* - s}{l - s}$$

$$= b_s + \frac{b_l - b_s}{l - s}(W_i^* - s) \qquad \square$$

The following lemma is a niche property of the optimal fractional solution constructed by the incremental bang-per-buck order algorithm (Appendix A.1) that can be easily derived and that we use in our proofs.

**Lemma 3.** *Let $i$ be the last advertiser that is allocated in the incremental bang-per-buck order algorithm and suppose it is allocated fractionally. Let $\mathbf{x}^* = \mathbf{x}(OPT)$ denote the optimal fractional allocation. Let $(b_s, s), (b_\ell, \ell)$ be the ads used in $x_i^*$. For all $j \neq i$, $\frac{b_\ell - b_s}{\ell - s} \leq \frac{b_j \cdot x_j^*}{W_j^*}$.*

*Proof.* Let $j$ be an advertiser with $j \neq i$. Since $j \neq i$, by Fact 1 allocation of $j$ in $\mathbf{x}^*$ is integral. Let $(b_{jk}, w_{jk})$ be the ad allocated to $j$. Let $(b_{jt}, w_{jt})$ be the ad that was previously allocated to $j$ when $(b_{jk}, w_{jk})$ was considered. Since the incremental bang-per-buck is defined by sorting ads in increasing order of their space, $w_{jk} \geq w_{jt}$. Thus $\frac{b_{jk} - b_{jt}}{w_{jk} - w_{jt}}$ is the incremental bang-per-buck that allowed $j$ to be selected and since $i$ is the last advertiser we have that $\frac{b_l - b_s}{l - s} \leq \frac{b_{jk} - b_{jt}}{w_{jk} - w_{jt}}$.

To conclude the proof, we show that $\frac{b_{jk} - b_{jt}}{w_{jk} - w_{jt}} \leq \frac{b_{jk}}{w_{jk}}$. This is true if and only if $\frac{b_{jk}}{w_{jk}} \leq \frac{b_{jt}}{w_{jt}}$. We have that $0 \leq w_{jt} \leq w_{jk}$. We can conclude that $0 \leq b_{jt} \leq b_{jk}$, as otherwise, $k$ is dominated by $t$. Finally, Suppose, $\frac{b_{jk}}{w_{jk}} > \frac{b_{jt}}{w_{jt}}$, then with simple rearrangement we can obtain that $\frac{b_{jk}}{w_{jk}} < \frac{b_{jt} - b_{jk}}{w_{jt} - w_{jk}}$ and $k$ is LP-dominated by 0 and $t$. $\square$

## A.2 GSP Pricing and Price of Anarchy

In Sponsored Search, a popular pricing choice is the Generalized Second Price (GSP) [Var07, EOS07]. While this is classically defined for a position auction with k ads being selected, it can be defined for any allocation algorithm that is monotone in the bid.

**Definition 4** (GSP). *For any allocation rule $x_i$ that is monotone in bid, and a bid profile $(\mathbf{b}, \mathbf{S})$, the GSP per click price for advertiser $i$ is the minimum bid below which the advertiser obtains a smaller amount of expected clicks: $cpc_i(\mathbf{b}, \mathbf{S}) = \arg\min_{b'_i : x_i(b'_i, S_i, \mathbf{b}_{-i}, \mathbf{S}_{-i}) = x_i(b_i, S_i, \mathbf{b}_{-i}, \mathbf{S}_{-i})} b'_i$. Given a GSP cost-per-click, the GSP payment is the cost-per-click times expected number of clicks: $p_i(\mathbf{b}, \mathbf{S}) = x_i(\mathbf{b}, \mathbf{S}) cpc_i(\mathbf{b}, \mathbf{S})$.*

The mechanism that charges GSP prices may not be truthful. We can study the Price of Anarchy (PoA) [KP99] to understand the effective social welfare. The notion of Price of Anarchy captures the inefficiency of a pure Nash equilibrium. Fix a valuation profile $(\mathbf{v}, \mathbf{A})$. A set of bids $(\mathbf{b}, \mathbf{S})$ is a *pure Nash equilibrium*, if for each $i$, for any $(b'_i, S_i)$, $u_i(v_i, A_i \to b_i, S_i; \mathbf{b}_{-i}, \mathbf{S}_{-i}) \geq u_i(v_i, A_i \to b'_i, S'_i; \mathbf{b}_{-i}, \mathbf{S}_{-i})$. The pure Price of Anarchy is the ratio of the optimal social welfare to the social welfare of the worst pure Nash equilibrium of the mechanism $\mathcal{M}$:
pure PoA $= \sup_{\mathbf{v}, \mathbf{A}, \text{ pure Nash}(\mathbf{b}, \mathbf{S})} \frac{SW(OPT(\mathbf{v}, \mathbf{A}))}{SW(\mathcal{M}(\mathbf{b}, \mathbf{S}))}$.

We will also consider Bayes-Nash Price of Anarchy. Let $(\mathbf{v}, \mathbf{A})$ be drawn from a (possibly correlated) distribution $\mathcal{D}$, and $D_i$ be the marginal of $i$ in $\mathcal{D}$. A *Bayes-Nash equilibrium* is a (possibly randomized) mapping from value, set of rich ads $(v_i, A_i)$ to a reported type $(b_i(v_i, A_i), S_i(v_i, A_i))$ for each $i$ and $(v_i, A_i) \in Support(D_i)$ such that, for any $b'_i$ and $S'_i \subseteq A_i$,

$$\mathbb{E}_{\mathbf{v}_{-i}, \mathbf{A}_{-i}, \mathbf{b}_{-i}, \mathbf{S}_{-i}}[u_i(v_i, A_i \to b_i, S_i; \mathbf{b}_{-i}, \mathbf{S}_{-i})] \geq \mathbb{E}_{\mathbf{v}_{-i}, \mathbf{A}_{-i}, \mathbf{b}_{-i}, \mathbf{S}_{-i}}[u_i(v_i, A_i \to b'_i, S'_i, \mathbf{b}_{-i}, \mathbf{S}_{-i})]$$

In the expression above the expectation is conditioned on $v_i$ and over random draws of $\mathbf{v}_{-i}, \mathbf{A}_{-i}$ and the competitors bids $\mathbf{b}_{-i}, \mathbf{S}_{-i}$ drawn from the mapping $b_j(v_j, A_j), S_j(v_j, A_j)$ for each $j \neq i$. The *Bayes-Nash Price of Anarchy (PoA)* is the ratio of the optimal social welfare to that of the worst Bayes-Nash equilibrium of the mechanism $\mathcal{M}$.

$$\text{Bayes-Nash POA} = \sup_{\mathcal{D}, (\mathbf{b}, \mathbf{S}) \text{ Bayes Nash eq.}} \frac{\mathbb{E}_{(\mathbf{v}, \mathbf{A})}[SW(OPT(\mathbf{v}, \mathbf{A}))]}{\mathbb{E}_{(\mathbf{v}, \mathbf{A}), (\mathbf{b}, \mathbf{S})}[SW(\mathcal{M}(\mathbf{b}(\mathbf{v}, \mathbf{A}), \mathbf{S}(\mathbf{v}, \mathbf{A})))]}.$$

For the PoA bounds we also focus on equilibria that satisfy the *no overbidding* condition, i.e. equilibria where each bid profile $\mathbf{b}, \mathbf{S}$ satisfies $b_i \leq v_i$ for every advertiser $i$. Note that, the equilibrium condition still allows for deviations that overbid. However, by the definition of GSP overbidding is dominated. If an advertiser can obtain a higher expected number of clicks by bidding higher than their value, then their GSP cost-per-click will be larger than their value $v_i$, and the advertiser obtains negative utility. On the other hand if the expected number of clicks is unchanged, then the GSP cost-per-click is also the same and the utility is unchanged as well, thus no advertiser will be able to gain by overbidding.

## B  Proofs from Section 3

*Proof of Lemma 1.* We note that, for a fixed set of rich ads $S'_i$ this is a single parameter setting (in the bids $b_i$). Since we use the same payment rule as Myerson, his result implies for any $b_i, S'_i$ $u_i(b_i, \mathbf{b}_{-i}, S'_i, \mathbf{S}_{-i}) \leq u_i(v_i, \mathbf{b}_{-i}, S'_i, \mathbf{S}_{-i})$. Thus the mechanism is truthful in bids.

Next we show that reporting a smaller set of rich ads and the true value is not beneficial, i.e. $u_i(v_i, S_i, \mathbf{b}_{-i}, \mathbf{S}_{-i}) \leq u_i(v_i, A_i, \mathbf{b}_{-i}, \mathbf{S}_{-i})$, for any $S_i \subseteq A_i$. From the definition of the payment, $u_i(v_i, S_i, \mathbf{b}_{-i}, \mathbf{S}_{-i}) = \int_0^{v_i} x_i(b, S_i, \mathbf{b}_{-i}, \mathbf{S}_{-i}) db$. Since the allocation rule is monotone, we have $x_i(b, \mathbf{b}_{-i}, A_i, \mathbf{S}_{-i}) \geq x_i(b, \mathbf{b}_{-i}, S_i, \mathbf{S}_{-i})$; the claim follows.

Finally we can chain these two results to show that misreporting both bid and the set of rich ads will also not increase a buyer's utility. Let $S_i$ be the reported set of rich ads, and fix $\mathbf{b}_{-i}$ and $\mathbf{S}_{-i}$. Recall that in our model the buyer can only report a subset of rich-ads, thus $S_i \subseteq A_i$. Further for any valid allocation rule $x_{ij} = 0$ for all $j \notin S_i$. Thus the utility when the true type is $A_i$ is equal to the utility when the true type is $S_i$: $u_i(\nu_i, A_i \to \nu'_i, S_i) = u_i(\nu_i, S_i \to \nu'_i, S_i)$ for all $\nu_i, \nu'_i$. Putting all the previous claims together we have:

$$\begin{aligned}
u_i(v_i, A_i \to v_i, A_i) &\geq u_i(v_i, A_i \to v_i, S_i) \quad \text{(By local IC)} \\
&= u_i(v_i, S_i \to v_i, S_i) \\
&\geq u_i(v_i, S_i \to b_i, S_i) \quad \text{(By local IC)} \\
&= u_i(v_i, A_i \to b_i, S_i) \qquad\qquad\qquad \square
\end{aligned}$$

*Proof of Theorem 1 continued.* Consider an instance with two advertisers $A, B$, and rich ads $\{(1, 1), (2 - \varepsilon, 2)\}$. The total space available is 3. The optimal allocation can randomly pick between $\{A : (1, 1), B : (2 - \varepsilon, 2)\}$ and $\{A : (2 - \varepsilon, 2), B : (1, 1)\}$, getting a total value of $3 - \varepsilon$. Therefore, in the output of any algorithm in this instance, some advertiser, say $B$, obtains value $x \le (3 - \varepsilon)/2$. A randomized monotone allocation rule must ensure that $B$'s value, if she hides $(1, 1)$, is at most $x$. In that case, the algorithm will randomize between $\{A : (1, 1), B : (2 - \varepsilon, 2)\}$ and $\{A : (2 - \varepsilon, 2)\}$, and it choose the first allocation with probability more than $\frac{x}{2-\varepsilon}$. The social welfare of this randomized allocation is at most $\frac{x}{2-\varepsilon} \cdot (3 - \varepsilon) + (1 - \frac{x}{2-\varepsilon}) \cdot (2 - \varepsilon) \le (11/12) \cdot (3 - \varepsilon)$. Recalling that OPT is $(3 - \varepsilon)$ concludes the proof. $\qquad\square$

# C    Proofs and Definitions from Section 4

## C.1    Formal description of $ALG_B$

**Algorithm 3** ($ALG_B$)**.** *Let $w$ denote the remaining space at any stage of the algorithm; initialize $w = W$. Let $E_i$ be the set of ads that are available to agent $i$ and let $\mathcal{E}$ denote the set $\cup_{i=1}^{n} E_i$; initialize $E_i = S_i$. Let $x_{ij}$ denote the fractional allocation of advertiser $i$ for the rich ad $j$. The total space allocated to advertiser $i$ is $W_i = \sum_{j \in \mathcal{M}} w_{ij} x_{ij}$.*

*While the remaining space $w$ is not zero:*

1. *Let $i$ be the advertiser whose rich ad $(b_{ij}, w_{ij})$ has the highest bang-per-buck, among all ads in $\mathcal{E}$. Let $(b_{ik}, w_{ik})$ be the ad previously allocated to $i$; $(b_{ik}, w_{ik}) = (0, 0)$, if no previous ad exists.*

2. *Remove all ads of $i$ (including ad $j$) with space at most $w_{ij}$ from $E_i$.*

3. *If $w \ge w_{ij} - w_{ik}$, add $w_{ij} - w_{ik}$ to the total space allocated to advertiser $i$, which now becomes $W_i = w_{ij}$. Allocate rich ad $j$ in that space, i.e. set $x_{ij} = 1$ (and $x_{ik} = 0$). Update $w = w - w_{ij} + w_{ik}$.*

4. *If $w < w_{ij} - w_{ik}$, add $w$ to the total space allocated to advertiser $i$, that is, $W_i = w_{ik} + w$. Allocate rich ad $j$ to $i$ fractionally with $x_{ij} = W_i/w_{ij}$ (and set $x_{ik} = 0$). Update $w = 0$.*

## C.2    Missing Proofs

*Proof of Theorem 2.* $ALG_B$ allocates some ads that are later replaced. We refer to such ads as temporarily allocated.

We first prove that an advertiser $i$ will not get allocated less space when bidding $b'_i > b_i$. For any agent $i \in \mathcal{N}$ and any $j \in S_i$, $ALG_B$ temporarily allocates $j$ if and only if the total space occupied by ads with higher bang-per-buck (counting only the largest such ad for each advertiser) is strictly less than $W$. For any ad $j \in S_i$, $j$'s bang-per-buck $b_{ij}/w_{ij}$ is increasing in the bid. Therefore, if $j \in S_i$ was temporarily allocated to $i$ when reporting $b_i$, then $j$ will definitely be temporarily allocated to $i$ under $b'_i > b_i$. If $j$ is the last ad to be allocated by the algorithm when reporting $b_i$ (and thus might not fit integrally in the remaining space under bid $b_i$), it will only be considered earlier under $b'_i > b_i$, and therefore the remaining space (before allocating $j$) can only be (weakly) larger. Thus, in all cases, the space allocated to $i$ under $b'_i$ is at least as much as under $b_i$.

We next show that by removing an ad $k \in S_i$ the space allocated to $i$ does not increase. Note that it is sufficient to prove monotonicity removing one rich-ad at a time. Monotonicity for removing a subset of rich-ads follows through transitivity. If no ad is allocated to $i$ under $S_i$, then definitely the same holds for $S_i \setminus \{k\}$. Otherwise, let $j \in S_i$ be the final ad allocated to $i$ under $S_i$. If $k$ was never (temporarily or otherwise) allocated under $S_i$, then the allocation under $S_i \setminus \{k\}$ remains the same. Therefore, we focus on the case that $k$ was allocated to $i$ under $S_i$.

First, consider the case that $k \ne j$. Let $\ell$ be an ad (temporarily or otherwise) allocated under $S_i \setminus \{k\}$, but not under $S_i$ (if no such ad exists, the claim follows). It must be that $b_{ik}/w_{ik} \ge b_{i\ell}/w_{i\ell}$ otherwise the algorithm under $S_i$ would have temporarily allocated $\ell$ before $k$. Note, if $w_{i\ell} > w_{ik}$ and the algorithm under $S_i$ did not allocate $\ell$ then it must be the case that space ran out before we got to $\ell$. This implies that, also under $S_i \setminus \{k\}$, the space will run out before we get to $\ell$. Hence $w_{i\ell} \le w_{ik}$. At the time when $\ell$ is allocated to $i$ (under $S_i \setminus \{k\}$), the total space allocated to bidders other than $i$

is weakly larger compared to the time when $k$ is allocated to $i$ (under $S_i$), while the space allocated to $i$ is weakly smaller.

Second, consider the case that $k = j$ and $i$ is not the last advertiser allocated by the algorithm. Removing $k$ leads to a different last ad, say $\ell$, for $i$ under $S_i \setminus \{k\}$. If this ad was temporarily allocated under $S_i$, the claim follows. Otherwise, $\ell$ must have a lower bang-per-buck than $k$. If $w_{i\ell} \leq w_{ik}$, the claim follows. Otherwise, all advertisers who got allocated after $i$ under $S_i$ (we know this set is non-empty) have the opportunity to claim a (weakly) larger amount of space under $S_i \setminus \{k\}$, before $\ell$ is considered. Thus, the maximum amount of space $i$ is allocated is $w_{ik}$.

Finally, suppose $k = j$, and $i$ is the last advertiser allocated by the algorithm. let $W_{-i} = W - w_{ij}x_{ij}$ be the space allocated to advertisers other than $i$, under $S_i$. Since they are allotted this space *before* ad $k$ is considered, the maximum space $i$ can get is $W - W_{-i}$, her allocation under $S_i$. $\qquad\square$

*Proof of Theorem 3 continued.* The following instance shows that the approximation of the algorithm is at least 3 (see Appendix C for the proof). Let $M$ be a large integer. There are four advertisers named $A, B, C, D$. We describe the set of rich ads, bang-per-buck and incremental bang-per-buck for each advertiser in Table 2. Total space $W = 2M - \varepsilon$. We assume ties are broken in the order $A, B, C, D$ (but the example can be constructed without ties).

| | rich ads | bpb | ibpb |
|---|---|---|---|
| $A$ | $(M, 1), (M + \varepsilon, M)$ | $M, \frac{M+\varepsilon}{M}$ | $M, \frac{\varepsilon}{M-1}$ |
| $B$ | $(1 + \varepsilon, 1), (M + \varepsilon, M)$ | $1 + \varepsilon, \frac{M+\varepsilon}{M}$ | $1 + \varepsilon, 1$ |
| $C$ | $(M - 1, M - 1)$ | $1$ | $1$ |
| $D$ | $(M + 2\varepsilon, 2M - \varepsilon)$ | $\frac{M+2\varepsilon}{2M-\varepsilon}$ | $\frac{M+2\varepsilon}{2M-\varepsilon}$ |

Table 2: Rich ads, Bang-per-buck(bpb) and Incremental Bang-per-bucks(ibpb) for each advertiser.

The fractional optimal solution can be constructed by allocating ads in the incremental-bang-per-buck order. The incremental bang-per-buck order is: $A : (M, 1), B : (1 + \varepsilon, 1), B : (M + \varepsilon, M), C : (M - 1, M - 1), D : (M + 2\varepsilon, 2M - \varepsilon), A : (M + \varepsilon, M)$. Any subsequent ad from the same advertiser fully replaces the previously selected ad. The allocation stops when the space runs out, so it will stop while allocating $C : (M - 1, M - 1)$ which will be allocated fractionally. The social welfare of the fractional optimal is $M + M + \varepsilon + \frac{(M-1-\varepsilon)}{M-1} \cdot (M - 1) = 3M - 1$.

$ALG_B$ considers ads in the order $A : (M, 1), B : (1 + \varepsilon, 1), A : (M + \varepsilon, M), B : (M + \varepsilon, M), C : (M - 1, M - 1), D : (M + 2\varepsilon, 2M - \varepsilon)$. Once again, any subsequent ad from the same advertiser fully replaces the previously selected ad. The algorithm stops when the space runs out. Thus the algorithm stops while allocating $B : (M + \varepsilon, M)$. $ALG_B$ will allocate space of $M$ to advertiser $A$ and space $M - \varepsilon$ to advertiser $B$. $ALG_I$ runs $ALG_B$ and post-processes to find the best ad for the allocated space. Thus $A$ is allocated $(M + \varepsilon, M)$ and $B$ is allocated $(1 + \varepsilon, 1)$. The social welfare of $ALG_I$ is $1 + \varepsilon + M + \varepsilon = M + 1 + 2\varepsilon$. The maximum value allocation will select $D : (M + 2\varepsilon, 2M - \varepsilon)$. Thus the expected value of the algorithm that randomly chooses between $ALG_I$ with probability $2/3$ and the largest value single ad with probability $1/3$ is $2/3(M + 1 + 2\varepsilon) + 1/3(M + 2\varepsilon) = M + 2/3 + 2\varepsilon$ and the ratio with the fractional optimal allocation is $\frac{3M-1}{M+2\varepsilon+2/3} = \frac{3-1/M}{1+\frac{2}{3M}+2\frac{\varepsilon}{M}}$. This ratio can be made arbitrarily close to 3 by choosing $\varepsilon = 1/M$ and $M$ that is large enough. $\qquad\square$

Now we prove all the claims from Section 4

*Proof of Observation 1.* We have that $\frac{b_{ij}}{w_{ij}} \geq \frac{b_{ij'}}{w_{ij'}}$ and $w_{ij} \geq w_{ij'}$. If we multiply these two inequalities we get: $b_{ij} \geq b_{ij'}$. If $w_{ij} = w_{ij'}$, then $j'$ is dominated by $j$. Otherwise, we will show that $j'$ is LP-dominated by $j$ and 0. We have that $0 < w_{ij'} < w_{ij}$ and $0 \leq b_{ij'} \leq b_{ij}$. It remains to show, $\frac{b_{ij}-b_{ij'}}{w_{ij}-w_{ij'}} \geq \frac{(b_{ij'})}{w_{ij'}}$. This is equivalent to $\frac{b_{ij}}{w_{ij}} \geq \frac{b_{ij'}}{w_{ij'}}$, which is true. $\qquad\square$

*Proof of Claim 1.* The post-processing step in $ALG_I$ integrally allocates the best ad that fits in $W_i$ for all advertisers $i \in \mathcal{N}$. Let $INT_i(w)$ denote best ad in $S_i$ that fits in space $w$.

For all $i \in \mathcal{I} \setminus \{i'\}$, since $W_i \geq W_i^*$ and the optimal allocation $\mathbf{x}_i^*$ is integral, we get $b_i \cdot \mathbf{x}_i(ALG_I) = b_i \cdot \mathbf{x}_i(INT_i(W_i)) \geq b_i \cdot \mathbf{x}_i(INT_i(W_i^*)) = b_i \cdot \mathbf{x}_i(OPT_i(W_i^*))$. Recall, by Lemma 2, we have $b_i \cdot \mathbf{x}_i(OPT_i(W_i^*)) = b_i \cdot \mathbf{x}_i^*$.

Thus we get $b_i \cdot \mathbf{x}_i^* \leq b_i \cdot \mathbf{x}_i(ALG_I)$ for all $i \in \mathcal{I} \setminus \{i'\}$. Further, by summing up the contributions of all $i \in \mathcal{I} \setminus \{i'\}$, we get $\text{Val}(\mathbf{x}^*, \mathcal{I} \setminus \{i'\}, \vec{W}^*) \leq \text{Val}(\mathbf{x}(ALG_I), \mathcal{I} \setminus \{i'\}, \vec{W})$. $\qquad\square$

*Proof of Claim 2.* $ALG_B$ consider ads in decreasing order of bang-per-buck, and moreover by Observation 1 it never "ignores" ads that are allocated in $OPT$. So, if $k \in \mathcal{K} \setminus \{i'\}$ had higher bang-per-buck in $OPT$ than any advertiser $i$ in $ALG_B$, then the space allotted for $k$ in $ALG_B$ would have been at least $W_k^*$; a contradiction. Note that this holds even if $i = i''$ is the last advertiser considered in $ALG_B$ (who potentially gets a fractional allocation). $\qquad\square$

*Proof of Claim 3.* From Lemma 2 we know that $b_\ell > b_s$ and $b_{i'} \cdot \mathbf{x}_{i'}^* = b_s + (b_\ell - b_s)\frac{W_{i'}^* - s}{\ell - s}$. Moreover if $(b_s, s)$ is not the null ad, i.e. $s > 0$, $\frac{b_\ell - b_s}{\ell - s} \leq \frac{b_\ell}{\ell} \leq \frac{b_s}{s}$,

Suppose that $i' \in \mathcal{K}$. If $s > W_{i'}$, then by the same argument as Claim 2 we have that for all $i \in \mathcal{N}$ the bang-per-buck of every ad in $ALG_B$ is at least $b_s/s \geq b_\ell/\ell \geq (b_\ell - b_s)/(\ell - s)$. If $s \leq W_{i'}$, we have $b_s \leq b_{i'} \cdot \mathbf{x}_i(ALG_I)$. And, by the same argument as Claim 2, we have that $\frac{b_\ell - b_s}{\ell - s} \leq \frac{b_\ell}{\ell} \leq \frac{b_j \mathbf{x}_j(ALG_I)}{W_j}$ for $j \in \mathcal{N}$.

Suppose that $i' \in \mathcal{I}$. Clearly, $s \leq W_{i'}^* \leq W_{i'}$, so we have $b_s \leq v_{i'} \cdot \mathbf{x}_{i'}(ALG_I)$. Let $k \in \mathcal{K}$ be some ad with $W_k^* > W_k$. By Lemma 3 we have $(b_\ell - b_s)/(\ell - s) \leq b_k \cdot \mathbf{x}_k^*/W_k^*$. Claim 2 gives us that $b_k \cdot \mathbf{x}_k^*/W_k^* \leq b_i \cdot \mathbf{x}_i(ALG_B)/W_i$ for all $i \in \mathcal{N}$. Thus we get $(b_\ell - b_s)/(\ell - s) \leq b_i \cdot \mathbf{x}_i(ALG_B)/W_i$ for all $i \in \mathcal{N}$. $\qquad\square$

### C.3 Computing Myerson Payments

Finally, we note that the truthful payment function matching our allocation rule (that gives the overall auction) can be computed in time that is polynomial in the number of advertisers and rich ads. Let $ALG_{max}$ be the algorithm which simply allocates the maximum valued ad. To compute the payment for an advertiser $i$, we need to compute the expected allocation as a function of $i$'s bid $b_i$. We can compute the expected allocation from $ALG_I$ and $ALG_{max}$ independently, and the final allocation is just $\frac{2}{3}x_i(ALG_I(b, \mathbf{b}_{-i}, \mathbf{S}) + \frac{1}{3}x_i(ALG_{max}(b, \mathbf{b}_{-i}, \mathbf{S})$. The payment is then $\frac{2}{3}p_i(ALG_I(\mathbf{b}, \mathbf{S})) + \frac{1}{3}p_i(ALG_{max}(\mathbf{b}, \mathbf{S}))$.

The payment for $ALG_{max}$ is simple: Advertiser $i$'s allocation is $\max_{j \in S_i} \alpha_{ij}$ if she is the highest value bidder, and zero otherwise. The expected payment $p_i$ is the second highest value $\max_{i' \neq i, j \in S_{i'}} b_{i'j}$ if $i$ is allocated and zero otherwise. The payment for $ALG_I$ can be computed by identifying possible thresholds for $i$'s bid where $i$'s allocation changes, and computing expected allocation for these thresholds. Advertiser $i$'s allocation can change *only* when his bang-per-buck for one of his ads is tied with that of another ad: there are therefore at most $O(n|\mathcal{S}|^2)$ such thresholds. Once we identify thresholds $t_1, t_2, ...$, the corresponding allocations can be computed by re-running the bang-per-buck allocation algorithm. Note that as $i$'s bid changes, the relative order of rich ads does not change for any advertiser $j \in \mathcal{N}$. Thus, the new bang-per-buck allocation can be computed in $O(n|\mathcal{S}|)$ time. Once we have the thresholds $t_0 = 0, t_1, t_2, ...$ and the corresponding expected allocations $x_i(ALG_I(t_j, b_{-i}, \mathbf{S}))$, the final payment is $\sum_{j=1} (x_i(ALG_I(t_j, b_{-i}, \mathbf{S})) - x_i(ALG_I(t_{j-1}, b_{-i}, \mathbf{S})) t_j$.

## D  Examples

The following example shows that $ALG_B$ might not be monotone.

**Example 4.** *Consider two advertisers $A$ and $B$. $A$ has two rich ads with (value,size) = $(2, 2)$ and $(1, 3)$, and $B$ has one rich ad with value size $(0.5, 3)$. Let the total space $W = 3$. $ALG_B$ will allocate $(1, 3)$. But if $A$ removed $(1, 3)$, then the algorithm allocates $(2, 2)$ to $A$ and $(0.5, 3)$ to $B$ fractionally.*

The following example shows that $ALG_I$ can be an arbitrarily bad approximation to the social welfare.

**Example 5.** *We have two advertisers A and B. A has one rich ad with (value, size) = $(\varepsilon, \varepsilon/2)$, and B has one rich ad $((M, M))$. The total space available is $W = M$.*

*Clearly, the optimal integer allocation is to award the entire space to B, to obtain social welfare $= M$. The fractional opt selects $A : (\varepsilon, \varepsilon/2)$ and $B : (M, M)$ with weight $M - \varepsilon/2$, obtaining social welfare $= M + \varepsilon/2$. We note that, in this instance the bang-per-buck algorithm also selects the fractional optimal allocation. At the same time, the integer allocation $\mathbf{x}(ALG_I)$, drops B, and only obtains social welfare $= \varepsilon$.*

# E  Price of Anarchy Bounds for GSP

In this section we prove bounds on the pure and Bayes-Nash PoA when monotone algorithms are paired with the GSP payment rule. Note that unlike the previous section, in this section we bound relative to the optimal integer allocation which we denote as $IntOPT$ .

We consider a mechanism $\mathcal{M}$ that runs $ALG_I$ with probability $1/2$ and allocates the maximum value ad with probability $1/2$. The corresponding GSP payment for either allocation rule is charged depending on the coin flip. The bidders utility is modelled in expectation over the random coin flip that selects between the two allocation rules. We refer to $ALG_I$ as bang-per-buck allocation and that of the maximum-value ad as the max-value allocation.

The following example illustrates that GSP paired with our allocation rules can be non-truthful.

**Example 6.** *Suppose there are two advertisers. Advertiser 1 has value $v_1 = 1$ and set of rich ads $A_1$ with one rich ad with (click probability, space): $(1/M, 1)$. Advertiser 2 has value $v_2 = 1$ and set of rich ads $A_2$ with two rich ads having (click probability, space): $(\varepsilon/M, 1), (1 + \varepsilon^2, M)$ respectively. Suppose total space $W = M$.*

*In this example, truth-telling is not an equilibrium. Suppose both advertisers bid truthfully. The bang-per-buck order is $2 : (1 + \varepsilon^2, M), 1 : (1/M, 1), 2 : (\varepsilon/M, 1)$. Thus advertiser 2 gets allocated $(1 + \varepsilon^2, M)$ and pays cost-per-click $\frac{1}{M} \cdot \frac{M}{1+\varepsilon^2}$, which is the minimum bid below which 2's bang-per-buck $(1 + \varepsilon^2) \cdot bid/M$ is lower than advertiser 1's. In the max-value allocation advertiser 2 is allocated $(1 + \varepsilon^2, M)$ with the GSP cost-per-click of $1/M \cdot 1/(1 + \varepsilon^2)$. Advertiser 2's utility is*

$$u_2(v_1, A_1, v_2, A_2) = \frac{1}{2}(1 + \varepsilon^2)\left(1 - \frac{1}{M} \cdot \frac{M}{(1+\varepsilon^2)}\right)$$
$$+ \frac{1}{2}(1 + \varepsilon^2)\left(1 - \frac{1}{M} \cdot \frac{1}{(1+\varepsilon^2)}\right) = \frac{1}{2}(1 + 2\varepsilon^2 - 1/M)$$

*However, if advertiser 2 bids $\frac{1}{2}$, advertiser 2's utility is*

$$u_2(1/2, A_1, v_2, A_2) = \frac{1}{2}(\varepsilon/M) + \frac{1}{2}(1 + \varepsilon^2 - 1/M)$$

*In the second case, the calculation for max-value is the same. In the bang-per-buck allocation, advertiser A has higher bang-per-buck and is allocated first. The rest of the space is allocated to advertiser 2 which is filled with the smaller ad. The GSP cost-per-click is zero. The latter utility is higher for $\varepsilon < 1/M$.*

We will use $SW_{bpb}$ and $SW_{max}$ to denote the social welfare of the bang-per-buck and max-value algorithms. Then, $SW_{\mathcal{M}}(\mathbf{b}, \mathbf{S}) = \frac{1}{2}SW_{bpb}(\mathbf{b}, \mathbf{S}) + \frac{1}{2}SW_{max}(\mathbf{b}, \mathbf{S})$. We use $u_i^{max}$ and $u_i^{bpb}$ to denote the utility of advertiser $i$ in the max-value and bang-per-buck allocation respectively.

The key challenge is in bounding the social welfare of the bang-per-buck allocation. Recall that the bang-per-buck algorithm allocates in the order of $b_{ij}/w_{ij}$. Rich ads from an advertiser that occupy less space than a previously allocated higher bang-per-buck rich ad are not picked. As the algorithm continues, it might replace a previously allocated rich ad [7] of an advertiser with another one that occupies more space (but may have lower value). The algorithm stops, when the next rich ad cannot fit within the available space or the set of rich ads runs out. We also post-process each advertiser to choose a rich ad with the highest value that fits within allocated space.

---

[7](We will refer to these ads as being temporarily allocated)

We will develop a little notation to make the argument cleaner. For each $i$ and $j \in S_i$ we use $(i,j)$ to denote this rich-ad. Without loss, we can assume that the space $w_{ij}$ occupied by any rich-ad $(i,j)$ is integer and the total space available $W$ is also integer. Let's think of the algorithm as consuming space in discrete units. For each unit of space, we can associate with it the rich ad that was first allocated to cover that unit of space. For the $k$'th unit of space, let $i(k)$ as the advertiser the $k$'th unit of space is allocated to, and $j(k)$ the rich ad $j(k) \in S_{i(k)}$ that was (temporarily) allocated for advertiser $i(k)$ when the $k$'th unit of space was first covered. Note that $j(k)$ may not be the final rich ad allocated to $i(k)$. In general, the algorithm stops before all the space runs out, in particular, because the next rich ad in the bang-per-buck order is too large to fit in the remaining space. We associate this rich-ad with each of the remaining units of space. It helps to be able to identify $(i(k), j(k))$ as we know that the bang-per-bucks $b_{i(k)j(k)}/w_{i(k)j(k)}$ are non-increasing as k increases.

We use the following lemma to relate an upper bound to the payment any "small" ad has to pay and the equilibrium social welfare.

**Lemma 4.** *Consider an equilibrium profile of per click bids $\mathbf{b}$ and sets of rich ads $\mathbf{S}$. We assume the bids satisfy no overbidding $b_i \leq v_i$ for each $i$. Let $k = \lfloor W/2 \rfloor + 1$, $(r,j) = (i(k), b(k))$, and $\beta = \frac{b_{rj}}{w_{rj}}$. Then, $\beta W \leq 2 \cdot SW_{bpb}(\mathbf{b}, \mathbf{S}) + 2 \cdot SW_{max}(\mathbf{b}, \mathbf{S})$.*

*Proof.* Recall that we defined $(i(k), j(k))$ as the rich-ad allocated by $ALG_B$ to the $k$'th unit of space. Then we let $k^* = \lfloor W/2 \rfloor + 1$, $(r,j) = (i(k^*), j(k^*))$ and $\beta = b_{rj}/w_{rj}$. Note that $(r,j)$ may not be allocated integrally if it is bigger than the remaining space.

Case (i): $(r,j)$ is allocated integrally
The rich-ads are also allocated contiguously. Let $K_0 = 0$ and $K_t = K_{t-1} + w_{i(K_t)j(K_t)}$. $K_t$ is the cumulative units of space covered by the first $t$ rich ads. For each $k \in \{K_{t-1}+1, K_{t-1}+2, \ldots, K_t\}$, $i(k) = i(K_t)$ and $j(k) = j(K_t)$. Then $\sum_{k=K_{t-1}+1}^{K_t} \frac{v_{i(k)j(k)}}{w_{i(k)j(k)}} = v_{i(K_t)j(K_t)}$.

$$
\begin{aligned}
\beta W \leq 2\beta k^* &= 2\frac{b_{i(k^*)j(k^*)}}{w_{i(k^*)j(k^*)}} k^* \\
&\leq 2 \sum_{k=1}^{k^*} \frac{b_{i(k)j(k)}}{w_{i(k)j(k)}} \\
&\leq 2 \sum_{k=1}^{k^*} \frac{v_{i(k)j(k)}}{w_{i(k)j(k)}} = 2 \sum_{t=1}^{} \sum_{k=K_{t-1}+1}^{K_t} \frac{v_{i(k)j(k)}}{w_{i(k)j(k)}} = 2 \sum_{t=1}^{} v_{i(K_t)j(K_t)} \\
&\leq 2SW_{bpb}(\mathbf{b}, \mathbf{S})
\end{aligned}
$$

In the first inequality, we use that $k^* > W/2$. Second inequality follows since, for each $k \leq k^*$, $b_{i(k)j(k)}/w_{i(k)j(k)} \geq \beta$. In the third step, we use the no-overbidding assumption. And the last inequality uses the fact that the temporary allocation $v_{i(K_i))j(K_i} \leq v_{i(K_i)\rho(i(K_i))}$ where $\rho(i)$ denotes the ad allocated to advertiser $i$ in the bang-per-buck allocation.
Case(ii) $(r,j)$ is not placed.
We have that $w_{rj} > W/2$ and $\beta \leq 2b_{rj}/W$. Hence,$\beta W \leq 2b_{rj} \leq 2SW_{max}(\mathbf{b}, \mathbf{S})$.

Combining the inequalities for the two cases, we have, $\beta W \leq 2SW_{bpb}(\mathbf{b}, \mathbf{S}) + 2SW_{max}(\mathbf{b}, \mathbf{S})$ □

The following lemma establishes a bound on the utility of an advertiser with rich ad $(i,t)$ of size less than $W/2$ bidding at least $\beta \frac{w_{it}}{\alpha_{it}}$ where $\alpha_{it}$ is the probability of click for rich ad $(i,t)$ . We use these deviations with the equilibrium condition to relate the social welfare of a bid-profile to that of a target optimal outcome.

**Lemma 5.** *Let $k = \lfloor W/2 \rfloor + 1$, $(r,j) = (i(k), b(k))$, and $\beta = b_{rj}/w_{rj}$ as defined above. Then for an advertiser $i$, with rich-ad $(i,t)$ with $w_{it} \leq W/2$ bidding $(y, \{t\})$ with $y = \min\{v_i, \beta w_{it}/\alpha_{it} + \varepsilon\}$, for some $\varepsilon$, $u_i^{bpb}(y, \{t\}, \mathbf{b}_{-i}, \mathbf{S}_{-i}) \geq v_{it} - \beta w_{it}$.*

We prove the following lemma, which covers Lemma 5, and is also used in the Bayes-Nash POA proof.

**Lemma 6.** *Let $k = \lfloor W/2 \rfloor + 1$, $(r,j) = (i(k), b(k))$, and $\beta = b_{rj}/w_{rj}$ as defined above. Then for an advertiser $i$, with rich-ad $(i,t)$ with $w_{it} \leq W/2$ bidding $(y, A_i)$ with $v_i \geq y \geq \beta \frac{w_{it}}{\alpha_{it}}$*

$$u_i^{bpb}(y, A_i, \mathbf{b}_{-i}, \mathbf{S}_{-i}) \geq v_{it} - y\alpha_{it}$$

*Moreover, for some $\varepsilon$, $y = \min\{v_i, \beta w_{it}/\alpha_{it} + \varepsilon\}$,*

$$u_i^{bpb}(y, \{t\}, \mathbf{b}_{-i}, \mathbf{S}_{-i}) \geq v_{it} - \beta w_{it}.$$

*Proof.* First, we will argue that with any bid $b_i' \in (\beta \frac{w_{it}}{\alpha_{it}}, v_i]$, and set of rich ads $S_i' \subseteq A_i$ such that $t \in S_i'$, $i$ will be allocated a rich ad of value at least $v_{it}$. This is tricky, because changing $i$'s bid also changes the allocation of all advertisers allocated earlier in the bang-per-buck order. However, as long as the space occupied by all ads other than $i$ when we get to $(i,t)$ is at most $W/2$, there is sufficient space remaining to place $(i,t)$. Since $b_i' > \beta \frac{w_{it}}{\alpha_{it}}$, $\frac{b_i' \alpha_{it}}{w_{it}} > \beta$, and $(i,t)$ appears before $(r,j)$ in bang-per-buck order. Hence the space allocated to others before $(i,t)$ is at most $W/2$. Thus the bang-per-buck algorithm will allocate at least $(i,t)$. Suppose the algorithm allocates $(i,j)$ instead, then $w_{ij} \geq w_{it}$ and the post processing step guarantees that $v_{ij} \geq v_{it}$ and $\alpha_{ij} \geq \alpha_{it}$. Since the GSP cost-per-click is at most the bid $b_i'$ we get

$$u_i^{bpb}(b_i', S_i', \mathbf{b}_{-i}, \mathbf{S}) \geq v_{ij} - b_i'\alpha_{ij} = \alpha_{ij}(v_i - b_i') \geq v_{it} - b_i'\alpha_{it}$$

In the last step we use that $b_i' \leq v_i$.

If $v_i < \beta \frac{w_{it}}{\alpha_{it}}$, let us consider the deviation $(b_i', \{t\})$ then by setting $b_i' = v_i$ we get $u_i^{bpb}(b_i', A_i, \mathbf{b}_{-i}, \mathbf{S}) \geq 0 \geq v_{it} - \beta w_{it}$. This is because the GSP cost-per-click is at most the bid $b_i' \leq \beta \frac{w_{it}}{\alpha_{it}}$.

Next we consider a deviation $b_i', \{(t)\}$ with $b_i' = \beta \frac{w_{it}}{\alpha_{it}} + \varepsilon < v_i$, for some $\varepsilon > 0$ such that $(r,j)$ immediately follows $(i,t)$ in bang-per-buck order. In this case $b_i'$ is still sufficient for $i$ to be allocated at least $(i,t)$. The GSP payment of $(i,t)$ is at most $\beta \cdot w_{it}$, as GSP payment is set by the bang-per-buck of $(r,j)$ or lower. Thus,

$$u_i^{bpb}(b_i', \{t\}, \mathbf{b}_{-i}, \mathbf{S}) \geq v_{it} - \beta w_{it} \qquad \square$$

Now we can prove the pure price of anarchy bound. In a pure PoA proof, we can consider deviations that depend on the other players bids which allows us to obtain a tighter analysis.

**Theorem 4.** *With the no-overbidding assumption, the pure PoA of the mechanism that selects using $ALG_I$ with probability $1/2$, selects the maximum value ad with probability $1/2$ and charges the GSP payment in each case is at most 6*

*Proof.* Consider the integer optimal allocation $IntOPT$. For each $i$, let $\tau(i)$ denote the rich ad selected for advertiser $i$ in $IntOPT$. If an advertiser $i$ is not allocated in $IntOPT$, we set $\tau(i) = 0$ which indicates the null ad. Let $(\mathbf{b}, \mathbf{S})$ denote a pure Nash equilibrium bid profile. The allocation of the mechanism is composed of two parts - bang-per-buck allocation and max-value-allocation. Let $\gamma(i)$ refer to the max-value allocation for advertiser $i$, but note that $\gamma(i) = 0$, i.e. the null ad, for all but one ad. Let $\rho(i)$ denote the rich ad allocated to advertiser $i$ in the bang-per-buck allocation.

First note that for any bid $b_i' \leq v_i$ and $S_i' \subseteq A_i$, $u_i^{bpb}(b_i', S_i', \mathbf{b}_{-i}, \mathbf{S}_{-i}) \geq 0$ and $u_i^{max}(b_i', S_i', \mathbf{b}_{-i}, \mathbf{S}_{-i}) \geq 0$. This is because the GSP cost-per-click is always less than the bid and the bid is less than value. Thus in either mechanism if the allocated rich ad is $(i,t)$, the utility $\alpha_{it}(v_i - cpc_i) \geq 0$.

We first bound the social welfare in the bang-per-buck allocation for advertisers that obtain an ad of space $\leq W/2$ in the optimal outcome $\tau$. Let $(r,j)$ be defined as in Lemma 5 and let $\beta = \frac{b_{rj}}{w_{rj}}$. By Lemma 5, for each $i$ with $w_{i\tau(i)} \leq W/2$, there exists an $\varepsilon$ such that with $b_i' = \min\{v_i, \beta \frac{w_{i\tau(i)}}{\alpha_{i\tau(i)}} + \varepsilon\}$, $u_i^{bpb}(b_i', \{\tau(i)\}, \mathbf{b}_{-i}, \mathbf{S}) \geq v_{i\tau(i)} - \beta w_{i\tau(i)}$. Thus we get,

$$(pv_{i\rho(i)} + (1-p)v_{i\gamma(i)}) \geq u_i(\mathbf{b}, \mathbf{S})$$
$$\geq u_i(b_i', \{\tau(i)\}, \mathbf{b}_{-i}, \mathbf{S})$$
$$\geq pu_i^{bpb}(b_i', \{\tau(i)\}, \mathbf{b}_{-i}, \mathbf{S})$$
$$\geq pv_{i\tau(i)} - p\beta w_{i\tau(i)}$$

Here, the first inequality uses the fact that the equilibrium payment is non-negative and the second is from the equilibrium condition. The third inequality follows from the fact that utility in max-value with GSP is non-negative. The last step is the estimation we have derived.

We have the above inequality for all $i$ such that $w_{i\tau(i)} \leq W/2$. Note that there can be at most one advertiser with $w_{i\tau(i)} > W/2$, denote this advertiser as $i^*$. Let $T = \mathcal{N} \setminus \{i^*\}$. Summing the above inequality over all $i \neq i^*$ we get,

$$
\begin{aligned}
pSW(T, \rho) + (1-p)SW(T, \gamma) &= \sum_{i \neq i^*} (pv_{i\rho(i)} + (1-p)v_{i\gamma(i)}) \\
&\geq \sum_{i \neq i^*} (pv_{i\tau(i)} - p\beta w_{i\tau(i)}) \\
&\geq pSW(T, \tau) - p \sum_{i \neq i^*} \beta w_{i\tau(i)}
\end{aligned}
\tag{3}
$$

**Case 1:** First we consider the case that there is no $i^*$ with $w_{i^*\tau(i*)} > W/2$. Starting from equation (3), we can bound $\sum_i w_{i\tau(i)} \leq W$ and use Lemma 4. We have $pSW_{bpb}(\mathbf{b}, \mathbf{S}) + (1-p)SW_{max}(\mathbf{b}, \mathbf{S}) \geq pSW(IntOPT) - 2p(SW_{bpb}(\mathbf{b}, \mathbf{S}) + SW_{max}(\mathbf{b}, \mathbf{S}))$. Setting $p = 1/2$ and rearranging, we get that the price of anarchy = $SW(IntOPT)/SW_{\mathcal{M}}(\mathbf{b}, \mathbf{S}) \leq 6$.

**Case 2:** If there is an ad $i^*$ with $w_{i^*\tau(i^*)} > W/2$. Then we have that $\sum_{i \neq i^*} w_{i\tau(i)} < W/2$. We will prove the following inequality for $i^*$.

$$
pv_{i^*\rho(i^*)} + (1-p)SW_{max}(\mathbf{b}, \mathbf{S}) \geq (1-p)v_{i^*\tau(i^*)}
\tag{4}
$$

Let $i^*$ deviate to bid truthfully. His utility on deviation,

$$
\begin{aligned}
u_{i^*}(v_{i^*}, A_{i^*}, \mathbf{b}_{-i^*}, \mathbf{S}_{-i^*}) &\geq (1-p)u_{i^*}^{max}(v_{i^*}, A_{i^*}, \mathbf{b}_{-i^*}, \mathbf{S}_{-i^*}) \\
&\geq (1-p) \max_{j \in A_{i^*}} v_{i^*j} - (1-p) \max_{i \neq i^*, j \in S_i} b_{ij} \\
&\geq (1-p)v_{i^*\tau(i^*)} - (1-p) \max_{i \neq i^*, j \in S_i} b_{ij}
\end{aligned}
\tag{5}
$$

Here the first inequality uses the fact that with bid less than value, the utility in the bang-per-buck allocation with GSP cost-per-click is non-negative. The second inequality is because $i^*$ may not be allocated when bidding $v_i^*$ in which case the competing bid is larger than $i^*$'s value.

If $i^*$ gets allocated in the max-value algorithm in equilibrium, $u_{i^*}(\mathbf{b}, \mathbf{S})$ is at most $pv_{i^*\rho(i^*)} + (1-p)v_{i^*\gamma(i^*)} - (1-p) \max_{i \neq i^*, j \in S_i} b_{ij}$. Using the equilibrium condition with $u_{i^*}(\mathbf{b}, \mathbf{S}) \geq u_{i^*}(v_{i^*}, A_{i^*}, \mathbf{b}_{-i^*}, \mathbf{S}_{-i^*})$, we get $pv_{i^*\rho(i^*)} + (1-p)v_{i^*\gamma(i^*)} \geq (1-p)v_{i^*\tau(i^*)}$. Equation (4) follows because $v_{i^*\gamma(i^*)} = SW_{max}(\mathbf{b}, \mathbf{S})$.

If $i^*$ does not get allocated in the max-value algorithm in the equilibrium, then $u_{i^*}(\mathbf{b}, \mathbf{S})$ is at most $pv_{i^*\rho(i^*)}$. Using $(1-p) \max_{i \neq i^*, j \in S_i} b_{ij} \leq (1-p)SW_{max}(\mathbf{b}, \mathbf{S})$, with the pure Nash equilibrium condition, and rearranging we get $pv_{i^*\rho(i^*)} + (1-p)SW_{max}(\mathbf{b}, \mathbf{S}) \geq (1-p)v_{i^*\tau(i^*)}$, i.e., equation (4).

Then adding equations (3) and (4), we get

$$
pSW_{bpb}(\mathbf{b}, \mathbf{S}) + 2(1-p)SW_{max}(\mathbf{b}, \mathbf{S}) \geq p(SW(IntOPT) - v_{i^*\tau(i^*)}) + (1-p)v_{i^*\tau(i^*)} - p\beta\frac{W}{2}
$$

where we use $SW_{bpb}(\mathbf{b}, \mathbf{S}) = SW(T, \rho) + v_{i^*\rho(i^*)}$, $SW_{max}(\mathbf{b}, \mathbf{S}) \geq SW(T, \gamma)$, $SW(IntOPT) = SW(T, \tau) + v_{i^*\tau(i^*)}$, and $\sum_{i \in T} w_{i\tau(i)} < W/2$. By Lemma 4, we have $\beta W/2 \leq SW_{bpb}(\mathbf{b}, \mathbf{S}) + SW_{max}(\mathbf{b}, \mathbf{S})$. Thus,

$$
pSW_{bpb}(\mathbf{b}, \mathbf{S}) + 2(1-p)SW_{max}(\mathbf{b}, \mathbf{S})) \geq pSW(IntOPT) - pSW_{bpb}(\mathbf{b}, \mathbf{S}) - pSW_{max}(\mathbf{b}, \mathbf{S}).
$$

Setting $p = 1/2$, we get that $6SW(\mathbf{b}, \mathbf{S}) \geq SW(IntOPT)$. So the price of anarchy is at most 6. □

**Bayes-Nash PoA** We also provide bounds on the Bayes-Nash Price of Anarchy when our allocation rule is paired with the GSP payment rule. Our proof technique is very similar to that of [CKK+15]. We borrow ideas from [CKK+15] and combine with techniques from our pure-PoA bound to prove a smoothness inequality, and obtain a bound on the Bayes-Nash PoA.

**Theorem 5.** *Under a no-overbidding assumption, the mechanism that runs $ALG_I$ with probability $1/2$ and allocates to the maximum valued ad with probability $1/2$, and charges the corresponding GSP price has a Bayes-Nash PoA of $\frac{6}{1-1/e}$.*

We will prove the bound using the smoothness framework. Our proof approach is similar to that of [CKK+15] for proving bounds on the price of anarchy of the GSP auction in the traditional position auction setting. However the knapsack constraint and the randomized allocation rule create unique challenges in our setting that we have to overcome.

We recall the definition of $(\lambda, \mu)$ semi-smoothness, as defined as [CKK+15], that extends [Rou15a], [NR10].

**Definition 5** ($(\lambda, \mu)$ semi-smooth games [CKK+15]). *A game is $(\lambda, \mu)$ semi-smooth if for any bid-profile $(\mathbf{b}, \mathbf{S})$, for each player $i$, there exists a randomized distribution over over bids $b'_i$ such that*

$$\sum_i \mathbb{E}_{b'_i(v_i)}[u_i(b'_i(v_i), b_{-i}, \mathbf{S})] \geq \lambda SW(OPT(\mathbf{v}, \mathbf{A})) - \mu SW(Alg(\mathbf{b}, \mathbf{S}))$$

The following lemma from [CKK+15] shows that smoothness inequality of the above form provides a bound on the Bayes-Nash POA.

**Lemma 7** ( [CKK+15]). *If a game is $(\lambda, \mu)$-semi-smooth and its social welfare is at least the sum of the players' utilities, then the Bayes-Nash POA is at most $(\mu + 1)/\lambda$.*

Thus, it only remains to prove the smoothness inequality. We prove that our mechanism is $(\frac{1}{2}(1-\frac{1}{e}), 2)$ semi-smooth, and hence obtain a Bayes-Nash POA bound of $6/(1 - \frac{1}{e}) \approx 9.49186$.

**Theorem 6.** *Under a no-overbidding assumption, the mechanism that runs $ALG_I$ with probability $1/2$ and allocates to the maximum valued ad with probability $1/2$, and charges the corresponding GSP price in each is $(\frac{1}{2}(1 - \frac{1}{e}), 2)$ semi-smooth.*

*Proof.* Fix a valuations profile $(\mathbf{v}, \mathbf{A})$ Consider the integer optimal allocation with valuation $(\mathbf{v}, \mathbf{A})$ as $OPT$. For each $i$, let $\tau(i)$ denote the rich-ad selected for advertiser $i$ in $OPT$. $\tau(i) = 0$ be the null ad with $\alpha_{i0} = 0$ and $w_{i0} = 0$ if advertiser $i$ is not allocated in $OPT$.

Let $(\mathbf{b}, \mathbf{S})$ denote a bid profile. The allocation of the mechanism is composed of two parts - bang-per-buck allocation and max-value-allocation. Let $\rho(i)$ denote the rich ad allocated to advertiser $i$ in the bang-per-buck allocation and $\gamma(i)$ denote the rich ad allocated to the advertiser $i$ in the bang-per-buck allocation. If an advertiser is not allocated we refer to the null ad with $\alpha_{i0} = 0$ and $w_{i0} = 0$. We will use $SW_{bpb}$ and $SW_{max}$ to denote the social welfare of the bang-per-buck and max-value allocation algorithms. Then,

$$SW_{\mathcal{M}}(\mathbf{b}, \mathbf{S}) = pSW_{bpb}(\mathbf{b}, \mathbf{S}) + (1-p)SW_{max}(\mathbf{b}, \mathbf{S}) = p\sum_i v_{i\rho(i)} + (1-p)v_{m,\mu}.$$

Most of the difficulty in proving the smoothness inequality is in reasoning about what happens in the bang-per-buck allocation. As in Lemma 4, let $k^* = \lfloor W/2 \rfloor + 1$ and $(r, j) = (i(k^*), j(k^*))$ be the rich-ad that would be allocated the $k^*$'th unit of space. Let $\beta = b_r \alpha_{rj}/w_{rj}$.

For any advertiser $i$, consider the advertiser deviates to bid $y$ drawn from a distribution on $[0, v_i(1 - 1/e)]$ with $f(y) = 1/(v_i - y)$. Then by Lemma 6, $u_i^{bpb}(y, A_i, \mathbf{b}_{-i}, \mathbf{S}_{-i}) \geq v_{i\tau(i)} - y\alpha_{i\tau(i)}$. If $\alpha_{i\tau(i)} \cdot y < \beta w_{i\tau(i)}$, we just lower bound the utility by 0.

$$\mathbb{E}_{y\sim F}[u_i^{bpb}(y, A_i, \mathbf{b}_{-i}, \mathbf{S}_{-i})] \geq \mathbb{E}_{y\sim F}[\alpha_{i\tau(i)}(v_i - y)I(\alpha_{i\tau(i)} \cdot y \geq \beta w_{i\tau(i)})]$$
$$= \int_0^{v_i(1-1/e)} \alpha_{i\tau(i)}(v_i - y)I(\alpha_{i\tau(i)} \cdot y \geq \beta w_{i\tau(i)}) \cdot \frac{1}{v_i - y}dy$$
$$= \alpha_{i\tau(i)}v_i(1 - 1/e) - \beta w_{i\tau(i)}$$

We have the above inequality for every $i$, with $w_{i\tau(i)} \leq W/2$.

**Case 1:** First consider the case where every advertiser has $w_{i\tau(i)} \leq W/2$ in OPT. Then we can sum over all $i$ and use the equilibrium condition to obtain a single smoothness inequality.

$$\sum_i \mathbb{E}_{y \sim f} u_i(y, A_i, \mathbf{b}_{-i}, \mathbf{S}_{-i})$$

$$= p \sum_i \mathbb{E}_{y \sim f} u_i^{bpb}(y, A_i, \mathbf{b}_{-i}, \mathbf{S}_{-i}) + (1-p) \sum_i \mathbb{E}_{y \sim f} u_i^{max}(y, A_i, \mathbf{b}_{-i}, \mathbf{S}_{-i})$$

$$\geq p \sum_i \mathbb{E}_{y \sim f} u_i^{bpb}(y, A_i, \mathbf{b}_{-i}, \mathbf{S}_{-i})$$

$$\geq p \sum_i [\alpha_{i\tau(i)} v_i (1 - 1/e) - \beta w_{i\tau(i)})]$$

$$\geq p(1 - 1/e) SW(OPT) - p\beta W$$

In the last step, we bound $\sum_i w_{i\tau(i)} \leq W$. If $p = 1/2$, by Lemma 4 we have $p\beta W \leq 2p(SW_{bpb}(\mathbf{b}, \mathbf{S}) + SW_{max}(\mathbf{b}, \mathbf{S}) = 2SW_{\mathcal{M}}(\mathbf{b}, \mathbf{S})$. And we have a smoothness inequality with parameters $(1/2(1 - 1/e), 2)$.

**Case 2:** Otherwise suppose $OPT$ has an advertiser with $w_{i\tau(i)} > W/2$. Note that OPT can have at most one advertiser with $w_{i\tau(i)} > W/2$. Denote this advertiser as $i^*$. We consider the utility of an advertiser $i^*$ as he deviates to bid $y$ drawn from distribution $f$ on $(0, v_{i^*}(1 - 1/e))$ with $f(y) = 1/(v_{i^*} - y)$. With a bid of $y$, the bidder will definitely be allocated in the max-value algorithm if $\alpha_{i^*\tau(i^*)} y \geq SW_{max}(\mathbf{b}, \mathbf{S})$. Note that this is a loose condition, $(i^*, \tau(i^*))$ may not be the most valuable rich ad for $i^*$, and if $i^*$ is allocated in the bid profile the threshold to win might be lower than $SW_{max}(\mathbf{b}, \mathbf{S})$ . But we will use this weaker condition. Again, recall that the GSP cost-per-click will be at most the bid $y$. Then we have,

$$\mathbb{E}_{y \sim f}[u_{i^*}^{max}(y, A_{i^*}, \mathbf{b}_{-i^*}, \mathbf{S}_{-i^*})] \geq \mathbb{E}_{y \sim f}[\alpha_{i^*\tau(i^*)}(v_{i^*} - y) I(\alpha_{i^*\tau(i^*)} y \geq SW_{max}(\mathbf{b}, \mathbf{S}))]$$

$$\geq \int_0^{v_{i^*}(1 - 1/e)} \alpha_{i^*\tau(i^*)}(v_{i^*} - y) \frac{1}{(v_{i^*} - y)} I(\alpha_{i^*\tau(i^*)} y \geq SW_{max}(\mathbf{b}, \mathbf{S})) dy$$

$$\geq \int_{\frac{SW_{max}(\mathbf{b}, \mathbf{S})}{\alpha_{i^*\tau(i^*)}}}^{v_{i^*}(1 - 1/e)} \alpha_{i^*\tau(i^*)} dy$$

$$\geq (1 - 1/e) \alpha_{i^*\tau(i^*)} v_{i^*} - SW_{max}(\mathbf{b}, \mathbf{S})$$

We can combine all the inequalities to obtain a single smoothness inequality.

$$\sum_i \mathbb{E}_{y \sim f} u_i(y, A_i, \mathbf{b}_{-i}, \mathbf{S}_{-i})$$

$$= p \sum_i \mathbb{E}_{y \sim f} u_i^{bpb}(y, A_i, \mathbf{b}_{-i}, \mathbf{S}_{-i}) + (1-p) \sum_i \mathbb{E}_{y \sim f} u_i^{max}(y, A_i, \mathbf{b}_{-i}, \mathbf{S}_{-i})$$

$$\geq p \sum_{i \neq i^*} \mathbb{E}_{y \sim f} u_i^{bpb}(y, A_i, \mathbf{b}_{-i}, \mathbf{S}_{-i}) + (1-p) \mathbb{E}_{y \sim f} u_{i^*}^{max}(y, A_{i^*}, \mathbf{b}_{-i^*}, \mathbf{S}_{-i^*})$$

$$\geq p \sum_{i \neq i^*} [\alpha_{i\tau(i)} v_i (1 - 1/e) - \beta w_{i\tau(i)})] + (1-p)[(1 - 1/e) \alpha_{i^*\tau(i^*)} v_{i^*} - SW_{max}(\mathbf{b}, \mathbf{S})]$$

$$\geq p(1 - 1/e) SW(OPT) - p\beta W/2 + (1 - 2p)(1 - 1/e) v_{i^*\tau(i^*)} - (1-p) SW_{max}(\mathbf{b}, \mathbf{S})$$

Here in the last step we bound $\sum_{i \neq i^*} w_{i\tau(i)} < W/2$.

By Lemma 4, we obtain an upper bound of $2SW_{bpb}(\mathbf{b}, \mathbf{S}) + 2SW_{max}(\mathbf{b}, \mathbf{S})$ on $\beta W$. Setting $p = 1/2$,

$$\sum_i \mathbb{E}_{y \sim f} u_i(y, A_i, \mathbf{b}_{-i}, \mathbf{S}_{-i}) \geq \frac{1}{2}(1 - \frac{1}{e}) SW(OPT) - \frac{1}{2}(SW_{bpb}(\mathbf{b}, \mathbf{S}) + SW_{max}(\mathbf{b}, \mathbf{S})) - \frac{1}{2} SW_{max}(\mathbf{b}, \mathbf{S}).$$

Recall that $SW_{\mathcal{M}}(\mathbf{b}, \mathbf{S}) = \frac{1}{2} SW_{bpb}(\mathbf{b}, \mathbf{S}) + \frac{1}{2} SW_{max}(\mathbf{b}, \mathbf{S})$. Hence, the smoothness inequality

$$\sum_i \mathbb{E}_{y \sim f} u_i(y, A_i, \mathbf{b}_{-i}, \mathbf{S}_{-i}) \geq 1/2(1 - 1/e) SW(OPT) - 2SW(\mathbf{b}, \mathbf{S}),$$

follows, and we obtain a bound on the Bayes-Nash POA of $6/(1 - 1/e) \approx 9.49186$. $\qquad \square$

# F Proofs from Section 5

We briefly sketch a proof that the new "heuristically better" algorithms are still monotone.

**Lemma 8.** *GreedyByBangPerBuck is monotone in both $b_i$ and $S_i$.*

*Proof.* Recall that in GreedyByBangPerBuck we do not stop immediately when encountering an ad that doesn't fit, but instead we will continue until we run out ads or knapsack space. We will refer to this as the bang-per-buck allocation. Finally, we do a post-processing step as usual.

Let $W_i$ denote the space allocated to agent $i$ before the post-processing step for each $i$. Since the post-processing step uses the space $W_i$ optimally, it is enough to show that the bang-per-buck allocation is space monotone. That is, we will show that $W_i$ is monotone in $b_i$ and $S_i$ (without worrying about the post-processing step). Also, wlog we can assume that the algorithm continues to consider ads in bang-per-buck order until we run out of ads (that is, if knapsack is full we will keep going without allocating anything else).

Clearly, for any $b_i' > b_i$, the advertiser will not get allocated lesser space when bidding $b_i'$. This is because, when reporting $b_i'$, all the ads of $i$ now have higher bang per buck (than compared to reporting $b_i$). So any ad $j \in S_i$ that was allocated (temporarily or otherwise) under $b_i$ will still be allocated in $b_i'$. In particular, the space available to $j$ is weakly higher under $b_i'$.

We next show that by removing any ad $j \in S_i$ the space allocated to $i$ does not increase. Note that, by transitivity, it is sufficient to prove monotonicity for removing one ad at a time. We see that by removing ads we can only increase the amount of space allocated to other advertisers. This is because the algorithm will continue until we run out of space. First, if $j$ was never (temporarily) allocated by the algorithm then either $j$ was dominated or there was not enough space to allocate $j$, in either case nothing changes under $S_i \setminus \{j\}$.

First consider the case where $j$ was the final ad allocated under $S_i$. This implies that either $j$ was the largest of $i$, or no ad $k \in S_i$ larger than $j$ (that is, $w_{ik} > w_{ij}$) had enough space available. In either case, by removing $j$ the space allocated to $i$ can only be lesser, since no such larger ad $k$ (if any) will have enough space available.

Next we consider the case where $j$ was temporarily allocated and finally a larger ad $k$ was allocated to $i$ instead. Then under $S_i \setminus \{j\}$ can only have weakly lesser space available to $k$. To see why this is true, consider "time step" at which $j$ was temporarily allocated. Note that $j$ being allocated does not reduce the space available to advertiser $i$ as subsequent rich ads of $i$ replace $j$ and can use the space previously allocated to $j$. If $w - w_{ij}$ is the space available to other agents at this time step, then by removing $j$ there is at least $w - w_{ij}$ space available for other agents at this time step (when $j$ is not available). Also while $j$ being allocated does cause the algorithm to drop other rich ads of lower bang-per-buck and lower size, since the size of these rich ads is necessary smaller they cannot make more space available under $S_i \setminus \{j\}$. Hence we see that the space allocated to $i$ is weakly lesser under $S_i \setminus \{j\}$.

□

**Lemma 9.** *GreedyByValue is monotone in both $b_i$ and $S_i$.*

*Proof.* Recall that in GreedyByValue we continue allocating ads in value order (while only allocating at most one ad per advertiser) until we run out of ads. That is, we skip past ads that do not fit in the available space and continue considering ads in value order. Once an agent $i$ get allocated we ignore all their remaining ads.

Suppose $i$ changes her bid from $b_i$ to $b_i' > b_i$. This implies, the value of all of $i$'s ads increased. Therefore, her allocation under $b_i'$ can only be better.

Similarly by dropping an ad $j \in S_i$ the allocation of $i$ can only be worse. If $j$ was not allocated under $S_i$ then nothing changes because either $j$ didn't fit (in which case we will still continue) or a better ad was allocated before $j$ and will be allocated under $S_i \setminus \{j\}$.

If $j$ was allocated, then $j$ was the highest valued ad of $i$ that fit. That is, all ads of $i$ with higher value than $j$ (if any) were considered before $j$ and did not fit. Thus they will still not be allocated even under $S_i \setminus \{j\}$.

| Algorithm | ApproxECPM | ApproxPayment | time-msec |
|---|---|---|---|
| GreedyByBPB-Myerson | 0.9493 | 0.66 | 3.2914 |
| GreedyByValue-Myerson | 0.9196 | 1.00 | 1.6087 |
| RandomizedGreedy-Myerson | $0.9393 \pm 0.0001$ | $0.7758 \pm 0.0007$ | $2.7316 \pm 0.0037$ |
| VCG | 1.0 | 1.0 | 30.8287 |

Table 3: Average performance of the mechanisms compared to VCG. We report average approximation of eCPM and payment relative to VCG and average running time in miliseconds. We report confidence intervals for the randomized mechanism.

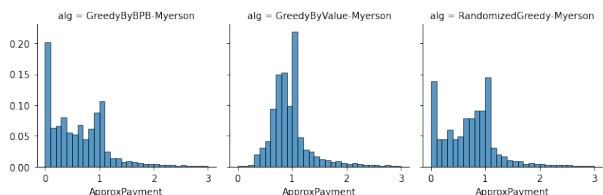

Figure 4: Histogram of ratio of revenue with MyersonPaymentRule for GreedyByValue, GreedyBy-BangPerBuck (GreedyByBPB) and RandomizedGreedy compared to VCG

$\square$

# G  Further Empirical Evaluation

**Comparison with Myerson Payment rule**    We compare the revenue performance of our allocation algorithms when paired with truthful payment rules.

We pair IntOPT with the VCG payment rule gives the VCG mechanism. The VCG payment for advertiser $i$ is computed by computing the brute-force integer OPT with the advertiser $i$ removed and subtracting from that the allocation of all advertisers other than advertiser $i$ in the optimal allocation. For our monotone mechanisms we compute Myerson payments as implied by Lemma 1. We compute the Myerson payment for advertiser $i$, by first computing the GSP cost per click at the submitted bid, setting a new-bid equal to GSP cost-per-click - $\epsilon$, and rerunning the allocation algorithm. This procedure is repeated until GSP cost-per-click or the allocation of advertiser $i$ is equal to zero.

We first compare the revenue performance of the four different truthful mechanisms that we have. Since these mechanisms are truthful, revenue can be compared without having to reason about equilibrium. Note however that we do not set reserve prices and reserve prices can be set and tuned differently to fully compare the revenue from these mechanisms. In Figure 4, the revenue from GreedyByBangPerBuck and GreedyByValue can be both higher and lower than VCG. GreedyBy-BangPerBuck tends to have lower revenue on average. This is probably due to the bang-per-buck allocation — large value ads might also occupy larger space and have lower bang-per-buck. Thus, even if a large value ad is used to price smaller ads that are selected, since the bang-per-buck is small the payment for the smaller ad is still small. We might also make better trade-off between revenue and efficiency by stopping the algorithms early.

In Figure 5, we compare the running time for computing truthful payments in each mechanism. We note that all the implementations can be further optimized and the choice of programming language can influence the running time as well. The results here are from mechanisms implemented in Python. We see that GreedybyBangPerBuck(GreedyByBPB) and GreedyByValue run much faster than VCG. The greedy allocation rules themselves are much faster than brute-force OPT (see Table 1), but the truthful payment rule computation for the Greedy algorithm requires more recursive calls to the Greedy allocation rule than that for VCG, this can be further optimized if required.

**Empirical results with cardinality constraint**    We also implemented our algorithm with the cardinality constraint. Suppose there is a limit of $k = 4$ distinct advertisers to be shown. This changes the optimization problem and the greedy-incremental-bang-per-buck algorithm no longer produces the optimal allocation. We compare the performance of simple greedy algorithms with Myerson payment rule with that of the VCG mechanism that computes the optimal allocation. The algorithm

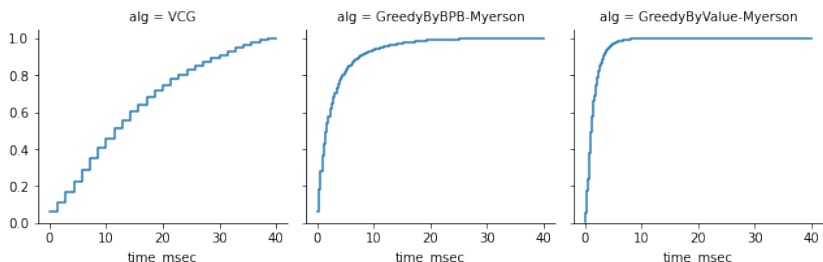

Figure 5: Histogram of ratio of running time in milliseconds for for GreedyByValue-Myerson, GreedyByBPB-Myerson and VCG mechanisms. We clip running time larger than 50 in the last bin.

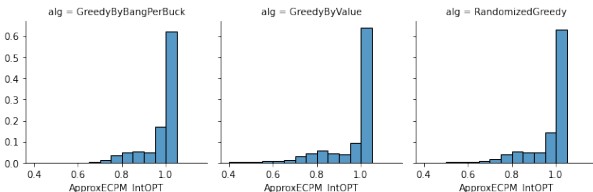

Figure 6: Approximation Factor of algorithms relative to IntOPT (a) Histogram of approximation factor for GreedyByValue, GreedyByBangPerBuck and randomized Greedy compared to IntOPT with a cardinality constraint of 4 ads

for computing optimal allocation recurses on subsets of ads and can be easily extended to track the cardinality constraint. The algorithm that allocates greedily by value of the rich-ad, will allocate the highest ad that fits within the available space and this algorithm can be stopped as as soon as $k$ distinct advertisers have been selected. To obtain the best social welfare using the greedy by bang-per-buck heuristic, more care is required. We cannot stop as soon as $k$ distinct advertisers are selected, instead we can improve social welfare further by replacing previously selected ads. Thus we extend our GreedyByBangPerBuck algorithm such that if the cardinality constraint is reached, it replace existing ad of the same advertiser if present (in this case the cardinality is unaffected) or replaces allocated ad of the advertiser that has the lowest value among all allocated advertisers.

In Figure 6, we compare the approximation factor of our greedy algorithms with cardinality constraint relative to the optimal integer allocation. We find that the worst-case approximation factors are still $0.6$ and $0.4$ for the GreedyByBangPerBuck and GreedyByValue algorithms, but $60\%$ of the queries have an approximation of $1.0$.