# OpenReview forum: "Simple Mechanisms for Welfare Maximization in Rich Advertising Auctions"
_NeurIPS.cc/2022/Conference — NeurIPS 2022 Accept_

### Official Review · Reviewer_j41M · 2022-07-09

**Rating:** 6
**Confidence:** 1
**Soundness:** 2 fair
**Presentation:** 2 fair
**Contribution:** 2 fair

**Summary:**

The paper study the internet as auctions problem with the advertiser has a rich set of ads with different value and space requirement. They adopt the `Myersonian'' approach and study allocation rules that are monotone both in the bid and set of rich ads.  The shows that the proposed a new, simple, greedy and monotone allocation rule that guarantees a  three approximation as compare the optimal.

**Questions:**

see weakness above.

**Strengths And Weaknesses:**

# Strengths
1. the study problem is interesting.
2. Theoretical results is provided.
3. Experimental results is conducted.

# Weakness
1. I am not familiar with this area. I am confused with the claim in the abstract that "We prove that, even though our monotone allocation rule paired with GSP is not truthful, its Price of Anarchy (PoA) is bounded.".  This seems create a contradiction.  Since they provide a monotone allocation rule in this paper  and also shows in  Lemma 1 that a monotone allocation with the corresponding payment rule will lead to truthful auction.  Will this lead to a truthful auction ?

2. The uploaded pdf version is obscure when zoom in.

Since I am not familiar with this area, I am willing to align my score with other reviewers if they are experts in this area.

---

> ### Author Response · Authors · 2022-07-29
> **Response to Reviewer j41M**
>
> Thank you for your review and constructive feedback. Below we answer your question.
>
> - “I am not familiar with this area. I am confused with the claim in the abstract that "We prove that, even though our monotone allocation rule paired with GSP is not truthful, its Price of Anarchy (PoA) is bounded.". This seems create a contradiction. Since they provide a monotone allocation rule in this paper and also shows in Lemma 1 that a monotone allocation with the corresponding payment rule will lead to truthful auction. Will this lead to a truthful auction?”
>
> We prove that a monotone allocation rule can be paired with an appropriate payment rule to obtain a truthful auction. However, not every payment rule leads to truthfulness. And, in our case, GSP is not the appropriate payment rule; the appropriate payment rule for truthfulness is stated in Lemma 1 (page 5, lines 229-230).
>
> For a similar example, consider the single item (i.e. single-parameter) setting and the monotone allocation rule “give the item to the agent with the largest bid”. Paired with the payment rule “if you get an item, pay the second highest bid” the overall auction is truthful. However, when paired with the payment rule “if you get an item, pay your bid”, the overall auction is not truthful.

---

### Official Review · Reviewer_f2B3 · 2022-07-11

**Rating:** 8
**Confidence:** 4
**Soundness:** 4 excellent
**Presentation:** 4 excellent
**Contribution:** 3 good

**Summary:**

The authors introduce a rich ads auction problem which can be considered as a special case of the well-known MULTI-CHOICE KNAPSACK problem. If the designer allocates ads using the bang-per-buck idea directly, there is an example that shows the allocation is not monotone. They show that no deterministic monotone rule can approximate the optimal welfare within factor 2. Their main result is a greedy allocation mechanism that approximates at least one-third of the optimal welfare.

**Questions:**

I have no questions.

**Limitations:**

There is no potential negative societal impact.

**Strengths And Weaknesses:**

There are some different versions of rich ad problems. The authors answer an open problem in [DSYZ10]. Both positive and negative results look strong to me. The mechanism is easy to implement in reality and the performance in the experiment is quite good. Overall, the paper is well-written.

No obvious weakness is found.

---

> ### Author Response · Authors · 2022-07-29
> **Response to Reviewer f2B3**
>
> Thank you for your review and constructive feedback.

---

### Official Review · Reviewer_rnSK · 2022-07-11

**Rating:** 7
**Confidence:** 5
**Soundness:** 4 excellent
**Presentation:** 3 good
**Contribution:** 4 excellent

**Summary:**

The basic model of search advertising auctions is by now well understood, but industry has long faced a practical challenge when adapting this model to rich formats that can show multiple versions of each add of different sizes and with different combinations of decorations.  The resulting combinatorial problem has been challenging from both computational and incentive perspectives.  This paper proposes a simple greedy approach which is a variant of the optimal fractional algorithm.  This forms the heart of a truthful mechanism which is shown to have both provable guarantees and strong empirical performance.

**Questions:**

Part of the reason these approaches are needed is the intractability of VCG, but in the experiments VCG seems faster than I would have expected (0.03 msec).  Is the data used for the experiment nicer in some way than average? If not why is this performance of VCG inadequate?

**Limitations:**

See weaknesses

**Strengths And Weaknesses:**

Strengths:
While the rich ad problem has received some prior attention in the literature, this is the first paper I have seen which presents what I would consider a largely “complete” solution.  The main theoretical result (Theorem 3) is compelling and supported by a number of other results which add to the richness of the paper.  I also appreciate that the analysis goes beyond simply achieving the theoretical bounds but discusses several heuristic improvements that do not affect the theory but improve performance in practice.  The empirical results provide a convincing demonstration of the benefits of the approach by showing strong performance relative to the theoretically intractable and practically 10x slower VCG.

Weaknesses:
- Results in the main paper feel somewhat cramped.  The paper does a reasonable job trying to fit things in within the page limits, but it does mean that quite a bit of the richness only really shows up in the appendix.
- As is sadly often the case in this space the reproducibility of the work is limited due to the commercial sensitivity of advertising datasets.  Nevertheless, some work has done a better job of at least defining models for generating synthetic data and running experiments on it as well to provide somewhat greater reproducibility.
- The one missing piece that led me to describe the solution as only “largely” complete is the omission of reserve pricing, which is crucial in practice.  This is acknowledged in Appendix G, and as mentioned the paper is already quite full, but I would have hoped for at least some discussion of how reserve prices can be reasonably applied since this doesn’t seem immediately obvious and I could imagine multiple ways to apply score-based reserves (raw score vs score-per-buck and per-advertiser vs per-variant).

---

> ### Author Response · Authors · 2022-07-29
> **Response to Reviewer rnSK**
>
> Thank you for your review and constructive feedback. We’d be happy to include a short discussion on reserve prices/revenue maximization in the main body of the final version of the paper. Below we answer your question.
>
> - “Part of the reason these approaches are needed is the intractability of VCG, but in the experiments VCG seems faster than I would have expected (0.03 msec). Is the data used for the experiment nicer in some way than average? If not why is this performance of VCG inadequate?”
>
> The data does contain some easy instances where the number of available advertisers is small and VCG runs very fast (See figure 4 in the appendix). While VCG is only 10 times slower on average, on the 50th percentile queries VCG can take upwards of 20 msec while it is less than 5 msec for our algorithm. Also note that the greedy allocation rule is very fast (See figure 6 in the appendix) and some of the latency comes from calculating the Myersonian payment. In short, Greedy with Myersonian or GSP payments is much faster than VCG. In practice, latency constraints are pretty tight and even queries in the tail need to be fast to provide a good experience to the end user.

---

> > ### Comment · Reviewer_rnSK · 2022-08-04
> > **Further clarification**
> >
> > I think Figure 4 in the appendix is more consistent with what I would have expected.  It has the median time for VCG between 10 and 20 msec.  But Table 1 lists the average (presumably mean) as 0.03 msec.  This is 3 orders of magnitude different!  So unless there is something quite different between the setups something feels off.  Or maybe Table 1 is actually reporting seconds?

---

> > > ### Author Response · Authors · 2022-08-04
> > > **Table 1 correction**
> > >
> > > The reviewer rnSK is indeed correct. The time reported in Table 1 is the mean time in seconds. We will correct it to be milli-seconds in the final version.

---

### Official Review · Reviewer_6VM1 · 2022-07-15

**Rating:** 4
**Confidence:** 2
**Soundness:** 3 good
**Presentation:** 2 fair
**Contribution:** 3 good

**Summary:**

In this paper, the authors consider the “Rich Advertising Auctions”, where advertisers can opt in or
out of showing different extensions with the ads, which also makes the setting not single-dimensional. The objective is to design truthful mechanisms to maximize social welfare.

Although the VCG mechanism is truthful and maximizes social welfare, it involves solving an NP-complete problem which is thus not efficient.

The first result set in this work is that the authors show no deterministic monotone rule can approximate the optimal welfare within a factor better than 2, and they design a simple greedy and monotone allocation rule that guarantees a 3 approximation. This algorithm combines a greedy procedure using the bang-per-buck order, and a randomizing between this procedure and the largest value ad.

The second result is to consider the Generalized Second Price (GSP) payment rule, which charges each advertiser the marginal threshold below which their allocation changes. The authors show that this mechanism is not truthful, but the resulting price of anarchy is bounded, under the standard assumption of no-overbidding.

The authors also provide an empirical evaluation of the designed mechanism on real-world data.


**Questions:**

Can you say a few more words about how to apply “Myersonion payment function” to the experiments?

**Strengths And Weaknesses:**

Strengths:

As far as I check, the results are correct. The paper is well written, and I can follow most parts easily. By the way, I actually think that most of the footnotes include quite important information for readers to understand the problem, which is not necessary to put in footnotes.

Another issue is about the presentation. I think some proofs in the main body can be sacrificed in exchange for a section of PoA, including the definitions there, like GSP and PoA. I think they are quite important for the authors to understand and appreciate the results.


Weaknesses:

I am not an expert on this work, and thus sought help from a colleague who works on Bayesian mechanism design. The comments are as follows. The new model is novel, which can be interesting to be studied. The results are not surprising, and most techniques are standard, including the Myerson Lemma, knapsack algorithm, and the way to analyze the price of anarchy. The results are good to be published, but may not be NeurIPS.

I kind of agree with the colleague, but with low confidence. Although I do not have enough knowledge of recent progress in this area, some key ideas used in this paper are indeed straightforward. For example, the randomization and the bang-per-buck order procedure are widely used in knapsack-related problems. The analysis of PoA is also not surprising if I did not miss the technical obstacles here; It bounds each advertiser’s utility gain from deviation but we need to carefully take knapsack constraint into consideration.

Overall, I think this paper is not exciting but can be accepted if space is allowed.

---

> ### Author Response · Authors · 2022-07-29
> **Response to Reviewer 6VM1**
>
> Thank you for your review and constructive feedback. Below we answer your questions.
>
> - “Can you say a few more words about how to apply “Myersonion payment function” to the experiments?”
>
> Let us clarify. By “Myersonian payment function” we mean the payment rule which, paired with our monotone allocation, gives an overall truthful auction (see Lemma 1, page 5, lines 229-230; and also note that this payment rule is very similar but not identical to the standard Myerson payment for single-parameter problems). We explain how to do this computation in Appendix G (lines 1012-1015). This computation is not as simple as computing the generalized second price (GSP), but, in the experiments, our auction is still much faster compared to VCG.
>
> - "The results are not surprising, and most techniques are standard, including the Myerson Lemma, knapsack algorithm, and the way to analyze the price of anarchy."
>
> Note that the underlying optimization problem is not knapsack but multi-choice knapsack, where the optimal fractional algorithm does not allocate in bang-per-buck order but instead allocates in incremental-bang-per-buck order. While the incremental-bang-per-buck algorithm does provide a 2-approximation, it is not monotone (which is necessary for our approach to give a truthful auction). We instead show that the bang-per-buck algorithm is monotone and that, mixing with the highest value ad, provides a good approximation to the optimal fractional allocation. Finally, the proof of this approximation guarantee (proof of Thm 3) is more involved than the 2-approximation for knapsack/multi-choice knapsack.
>
> For the Price of Anarchy result, indeed the blueprint of the proof (smoothness) is standard. However, the difficult part, bounding the utility after deviation, must be done from scratch for every given setup/algorithm. And, because of the randomization in our algorithm, getting such bounds is further challenging here.

---

### Author Response · Authors · 2022-07-29
**Response to all reviewers**

We would like to thank all reviewers for their constructive feedback. We will incorporate the valuable suggestions from all reviewers in the final version of this paper. Since there are no concerns/questions that are common across reviewers we answer each reviewer’s questions in separate, independent responses below.

---

### Meta-Review · Area_Chair_2NDz · 2022-08-26

**Recommendation:** Accept
**Confidence:** Certain

**Metareview:**

Reviewers agreed that rich ad auction is significant and are excited about theoretical bounds on the positive result (achieved by a simple mechanism) and the negative result. Overall, this is a solid theoretical paper on an important and classical problem in industry.

**Award:**

No

---

### Decision · Program_Chairs · 2022-09-14

Accept